# Recent Trends in the Development of Novel Metal-Based Antineoplastic Drugs

**DOI:** 10.3390/molecules28041959

**Published:** 2023-02-18

**Authors:** Lozan Todorov, Irena Kostova

**Affiliations:** Department of Chemistry, Faculty of Pharmacy, Medical University-Sofia, 1504 Sofia, Bulgaria

**Keywords:** platinum, ruthenium, gallium, gold, lanthanum, coordination complexes, oncology, cancer research

## Abstract

Since the accidental discovery of the anticancer properties of cisplatin more than half a century ago, significant efforts by the broad scientific community have been and are currently being invested into the search for metal complexes with antitumor activity. Coordination compounds of transition metals such as platinum (Pt), ruthenium (Ru) and gold (Au) have proven their effectiveness as diagnostic and/or antiproliferative agents. In recent years, experimental work on the potential applications of elements including lanthanum (La) and the post-transition metal gallium (Ga) in the field of oncology has been gaining traction. The authors of the present review article aim to help the reader “catch up” with some of the latest developments in the vast subject of coordination compounds in oncology. Herewith is offered a review of the published scientific literature on anticancer coordination compounds of Pt, Ru, Au, Ga and La that has been released over the past three years with the hope readers find the following article informative and helpful.

## 1. Introduction

Oncological diseases are the second leading cause of death worldwide with 9.6 million deaths in 2018 as per the World Health Organization fact sheet. Cancer treatment involves surgery, chemotherapy and radiotherapy [1]. Novel photodynamic therapy is a method that involves administration of a photosensitizing compound, followed by irradiation with an absorbance maximum wavelength and subsequent, localized formation of reactive oxygen species (ROS), such as singlet oxygen [2]. The aim is to produce significant, toxicological effects in the area of the tumor tissue, causing apoptosis, necrosis and finally—cell death [3]. Transition metals play an essential role in the chemistry of life. Serving as cofactors in enzymatic active sites, they enable the great multitude of selective catalytic conversions necessary for maintaining biological processes [4]. In living organisms, transition metals can be found in trace amounts. Excessive intake of such elements can cause a variety of toxicological effects, including carcinogenesis [5]. Yet exactly that toxicological potential presents the fundament of transition-metal-based anticancer therapies [6]. Intracellular release of toxic metal ions such as Pt(II)/(IV), Au(I)/(III), Ru(II)/(III), La(III), Ga(III) and many others has been a staple of metal-based anticancer treatments ever since the discovery of cisplatin’s antiproliferative properties more than half a century ago [7]. The authors of the present review aim to inform the reader about developments over the past three years in the research of Pt, Ru, Au and La transition metal complexes with potential antitumor properties. Gallium as a post-transition metal is also discussed since the biological activity of its ion is well-established [8] and the search for novel gallium complexes with anticancer action is gaining traction once again [9,10].

## 2. Novel Metal Complexes in Cancer Therapy

### 2.1. Platinum Coordination Compounds

Platinum-based drugs are widely used in anticancer chemotherapy. The first coordination complex discovered with clinically viable antitumor action was *cis*-diamminedichloroplatinum(II) ubiquitously known as cisplatin. Its biological activity was serendipitously discovered in the 1960s by Rosenberg and coworkers [11]. Together with carboplatin, oxaliplatin, nedaplatin and lobaplatin it is a choice drug for treatment of a variety of malignancies such as testicular, colorectal, ovarian and breast cancer [12]. Cisplatin is characterized by low specificity and its clinical use is therefore associated with systemic toxicities [12]. Carboplatin is considered a second-generation platinum-based drug with a higher degree of biosafety [13] that allows for treatment with higher dosages. The main mechanism of action of Pt(II) involves DNA binding, forming intra-strand and inter-strand crosslinks, changing DNA structure, resulting in cell cycle arrest and apoptosis in actively proliferating tumor cells [14]. Platinum drug resistance is a significant issue in chemotherapy. In order to be overcome, a third-generation complex, oxaliplatin, was introduced in 1996 [15]. The mechanisms of action of these drugs, their toxicities and modes of development of drug resistance are well-known and described in detail in the scientific literature [16,17], and a number of exhaustive reviews on the subject have been published [6,18,19]. An alternative area of research when it comes to platinum-based anticancer drugs involves Pt(IV) coordination compounds. Compared to Pt(II) they have higher coordination numbers (6 vs. 4), improved stability and reduced side effects. The ability to coordinate axial ligands allows for greater structural modification. Additionally, within the intracellular medium they are reduced to their active Pt(II) counterparts, with the axial ligands leaving. For these reasons, Pt(IV) complexes are viewed as potential platinum prodrugs [20,21]. A very attractive and informative review on the subject of Pt(IV) in cancer research [21] presents exciting results both in terms of safety and drug resistance. Notably, coordinating biologically active ligands could produce “multi-action” and “cancer-seeking” Pt(IV) prodrugs with enhanced efficacy and multiple mechanisms of action [21]. Promising results have been observed in vitro when treating cancer cell lines; however, there is a long way to go until safety and efficacy in humans would be properly assessed. Below are introduced the most recent Pt(II) and Pt(IV) complexes discussed in the scientific literature over the past three years.

Zhu and coworkers tested a variety of mitochondria-targeted platinum complexes for anticancer activity against lung cancer [22]. They modified pyriplatin with a triphenylphosphonium moiety (Figure 1) with the aim to selectively target and penetrate the inner mitochondrial membrane.

Three complexes were synthesized with a triphenylphosphonium substitute, attached at the ortho-, meta- and para-positions of the pyridine ring. They were tested for antiproliferative activity against lung cancer A549, cervical cancer HeLa, hepatocellular carcinoma SMMC cells and normal liver HL-7702, with the aid of the 3-(4,5-dimethylthiazol-2-yl)-2,5-diphenyltetrazolium bromide (MTT, 48 h) assay. The ortho-substituted complex manifested the highest anticancer cytotoxicity (IC_50_ = 8.7 μM against A549), even compared to the positive standards cisplatin and pyriplatin. Its IC_50_ against healthy HL-7702 cells was about six times higher (IC_50_ = 64.5), speaking to the improved selectivity of the compound. Tumor growth in mice, implanted with A549, was significantly suppressed. Pt was discovered to accumulate primarily in liver and kidneys. More Pt was accumulated in the lungs and tumor tissue, compared to cisplatin. After 24 h of cultivation the ortho-substituted complex was found to produce mitochondrial DNA (mtDNA) lesions, reduce oxygen consumption and decrease dramatically (by 87%) the mitochondrial membrane potential in A549 cells. Cellular uptake was observed to increase with lipophilicity.

Eskandari et al. [23] synthesized a triangular polynuclear complex, containing three Pt(II) centers and tested it against cancer stem cell (CSC)-enriched human mammary epithelial cells HMLER-shEcad and CSC-depleted HMLER lines (MTT assay, 72 h). Mono- and dinuclear analogs were also tested. Notably, the more Pt(II) centers in the complex, the greater the antiproliferative activity. The trinuclear complex had the lowest IC_50_ (2.24 μM against HMLER and 1.26 against HMLER-shEcad). Its activity was greater than the positive controls cisplatin, carboplatin and salinomycin. The IC_50_ against non-malignant breast MCF10A cells was about two times greater than against HMLER-shEcad, showing good selectivity. It manifested significant non-covalent DNA-intercalative and groove-binding activity.

He et al. [24] synthesized a folate-containing Pt(II) complex (Figure 2) with the idea to specifically target folate receptors that tend to be overexpressed on the surfaces of breast cancer cells.

After 72 h’s exposure, the complex significantly suppressed MCF-7 cell viability (IC_50_ = 87) compared to the negative control. The complex increased Bak1/Bclx ratios after 24 h of incubation, compared to cisplatin, a possible sign of pro-apoptotic activity. Caspase-3 activity was also increased.

Adams and coworkers synthesized a number of Pt(II)-terpyridine complexes (Figure 3) and subsequently tested them for in vitro antiproliferative properties against colon cancer HCT 116, colon adenocarcinoma SW480, lung cancer NCI-H460 and endometrial carcinoma SiHa cell lines (72 h incubation) [25]. 

Substituting the hydroxyl group of complex 1 with an organic mustard-type side chain significantly increased the antiproliferative activity of the complex against all cell lines tested. IC_50_ values decreased in the following order complex 3 (IC_50_ between 0.4 and 4.0) > complex 2 (IC_50_ between 1.0 and 5.6) > complex 1(IC_50_ between 14 and 22). What should be noted is that the corresponding ligands themselves were very active in the nanomolar range. Adding Pt(II) to form complexes increased IC_50_ values more than ten-fold. Despite that, complexes 2 and 3 suppressed cancer cell growth to the same degree or even further than the positive control cisplatin. Additional assays showed rapid binding to l-histidine, 9-ethylguanine and l-cysteine.

Another series of four platinum–terpyridine complexes [26] were studied as promising antiproliferative agents against A549, its cisplatin-resistant subline A549/DDP, epidermoid carcinoma A431, HeLa and MCF-7. Results showed that all tested substances were as active, or more active, compared to the positive control cisplatin, targeting not only DNA, but also membrane proteins. Epidermal growth factor receptor (EGFR) inhibitory activity was estimated to be higher (IC_50_ about 10 μM for the complexes) than that of gefitinib (IC_50_ of about 90 μM). 

Kutlu and coworkers [27] synthesized two pyridine-based Pt(II) complexes (Figure 4) and tested them against the colon cancer cell line (DLD-1). After 24 h incubation, applying the MTT assay, complex 1 behaved as a stronger cytostatic (IC_50_ = 25.79 μM), compared to complex 2. 

The presence of electron-withdrawing functional groups (such as fluorine atoms in complex 1) was deemed responsible for the increased antiproliferative effect. In this case, the authors proposed that the bond between the pyridine nitrogen and the platinum ion decreases due to lower electron density, causing an effective increase in interaction with DNA. Electron donors such as the amino- and methyl- groups in complex 2 would tend to have the opposite effect.

Mbugua and coworkers [28] synthesized Pt(II) and Pd(II) complexes with pyrrole-substituted Schiff bases. The newly-generated substances were tested with the aid of the MTT test (24 h exposure) against the colorectal adenocarcinoma Caco-2, HeLa, Hep-G2, MCF-7 and bone cancer PC-3 cell lines. Non-cancerous MCF-12A was also tested. The most active Pt(II) complex (Figure 5) was extremely potent against Hep-G2, with IC_50_ being in the nanomolar range (IC_50_ = 0.3 μM). It also suppressed the Caco-2, MCF-7 and PC-3 (IC_50_ ranging between 16 and30 μM).

The complex manifests a strong DNA-binding ability, the authors proposed using it as a possible viable candidate for DNA intercalation. Its low toxicity against the healthy MCF-12A shows its potential as a selective molecule with anticancer activity.

Another Schiff base containing complex was synthesized, characterized and tested for antiproliferative activity against MCF-7, Hep-G2, HeLa and A549 cancer cells and non-cancerous NHDF cells [29]. The complex exhibited pronounced activity against MCF-7. Its antiproliferative behavior against the cancer cell lines was significant, though lower than that of cisplatin. On the other hand, toxicity against the healthy NHDF cells was low, showing better selectivity than the positive control. Additionally, the ligand itself was less active than the complex. Significant calf thymus DNA (CT-DNA) binding was observed.

A number of potent phenanthriplatin analogs [30] (Figure 6) with multidentate ligands have been synthesized and tested (MTT assay, 72 h) for anticancer activity against the ovarian cancer A2780, its cisplatin-resistant variant A2780cis, ovarian adenocarcinoma SKOV-3, triple-negative breast cancer MDA-MB-231 and A549 cancer cell lines. Toxicity against normal MET5A and HEK-293 cells was also measured.

Both ligands were inactive in the absence of the Pt(II) coordination center. Complex 2 was much more effective than complex 1. After 72 h incubation, complex 2 was found to be more potent than cisplatin against all cancer cell lines, including the cisplatin-resistant A2780cis (IC_50_ = 0.55 μM). Its toxicity against normal cells was decreased compared to phenanthriplatin. Cellular uptake was also improved and apoptosis was induced.

Tham and coworkers [31] followed a novel approach in terms of Pt-based anticancer therapy. They attempted to cause immunogenic cell death by causing endoplasmic reticulum stress. The novel compounds were tested against CT26 colorectal carcinoma (IC_50_ between 1.5 and 8.8 μM). Increasing lipophilicity caused a corresponding increase in cellular uptake. The most potent complex increased intracellular ROS levels and caused endoplasmic reticulum stress. An induction in phagocytosis-related signaling was observed.

Derivatives of benzothiazole aniline [32] (a substance with known anticancer properties) were used as ligands it order to produce selective anticancer agents (Figure 7). Cell viability was measured using the cell counting kit 8 (CCK-8) method (24 h cultivation).

All three complexes were less active than their corresponding organic ligands, but more active that benzothiazole aniline itself. Complexes 1 and 2 were particularly effective against the Hep-G2 human hepatic carcinoma cell line (half-maximal inhibitory concentration below 30 μM). Complex 1 moderately suppressed the proliferation of rat glioma C6, HeLa, colorectal adenocarcinoma HT-29 and MCF-7 cancer cell lines (IC_50_ below 100 μM). Toxicity against a variety of non-malignant cell lines was measured, revealing good selectivity toward cancer cells. Molecular docking revealed that complex 1 acted as an intercalating agent, binding to the minor groove of DNA.

A series of pyridine co-ligand functionalized cationic complexes inhibited proliferation in MCF-7, A549 and Hep-G2 cancer cells [33]. MTT testing proved a concentration-dependent antiproliferative effect (48 h treatment). The most active complexes are displayed on Figure 8.

Cancer cell tests revealed complexes 1 and 2 to be more potent than the positive control oxaliplatin and as potent as cisplatin. Complex 3 was more potent than both positive controls. Clonogenic studies showed that these three complexes suppressed the clonogenic potential of the tested cell lines. Cancer cell migration, invasion and cancer stem cell spheroid formation were decreased. Complex 3 demonstrated potential to target sterol regulatory element-binding protein 1 (SREBP-1)-dependent signaling pathways, thus inhibiting lipid biogenesis.

Nadar and coworkers [34] designed radioactive platinum(II)–bisphosphonate complexes in an attempt to specifically target bone cancer. The aim was to combine the anticancer activity of Pt(II) with the good radionuclide potential of its radioactive isotope ^195^Pt and the bone-targeting properties of bisphosphonates. This novel approach yielded promising results as Pt uptake, due to treatment of mice with the novel complex, was concentrated in the hard tissues, compared to a Pt(II) bearing, non-bisphosphonate positive control.

Pt(II), liganded with two bidentate analogs of thiourea (a known anticancer molecule) exhibited a moderate antiproliferative effect (MTT assay, 48 h) against colorectal cancer LoVo and MCF-7 with an IC_50_ greater than 100 μM [35]. Interestingly, if Pt(II) was exchanged with Pd(II) as a coordination center, then antiproliferative activity dramatically improved (IC_50_ between 10.44 and 62.86 μM), approaching that of cisplatin.

Mononuclear, Schiff base macrocyclic ligands were synthesized and coordinated with Pt(II) [36] (Figure 9). MTT assay (24 h incubation) was performed in order to estimate potential antiproliferative activity. Both ligands significantly suppressed the HeLa (IC_50_ between 12 and 15 μM) and A549 cell lines (IC_50_ = 10 μM). The addition of Pt(II) further improved the observed effect with an IC_50_ between 6 and 11 μM.

A Pt(II) complex with an ONN-“pincer” ligand [37] showed significant antiproliferative effect against Hep-G2 cells (IC_50_ = 6 to 12 μM), comparable to cisplatin. Activity against normal hPBMC was very mild, where IC_50_ > 200 μM.

Yambulatov and coworkers have synthesized a series of Pt(II) complexes with substituted 1,4-diaza-1,3-butadienes (redox-active, “non-innocent” ligands) [38]. They were tested for antiproliferative activity against malignant SKOV-3 and normal HDF cell lines. One of the complexes manifested significant activity against SCOV-3, similar to cisplatin (IC_50_ = 12 μM) and high toxicity against the non-cancerous cell line (IC_50_ = 9 μM)

Octahedral platinum (IV) complexes with the non-steroidal anti-inflammatory drugs indomethacin and acetylsalicylic acid as axial ligands have been reported [39]. After 72 h of incubation, the MTT assay showed significant cytotoxicity against a large panel of cancer cell lines. Particularly notable was the manifested activity against ovarian A2780 and cisplatin-resistant ovarian (ADDP) cancer cell lines. Activity against HT-29 colon cancer was also significantly increased in comparison to the positive controls cisplatin, carboplatin and oxaliplatin.

Another Pt(IV) prodrug was designed with two axial maleimide moieties to facilitate albumin binding [40]. Testing was performed on an in vivo murine cancer CT26 model. Twenty days after the beginning of treatment, tumor growth was significantly suppressed compared to the positive control oxaliplatin. Survival of the test animals was prolonged, some even entered remission and one was completely cured. The authors propose the formation of a stable albumin adduct in the blood stream, followed by cellular endocytosis and subsequent reduction of Pt(IV) to active Pt(II).

A series of mono-axial octahedral diazido Pt(IV) complexes with coumarin 3-carboxylate (an anticancer agent), 4-phenylbutyrate or dichloroacetate (PDK inhibitors) and their diaxial functionalized analogs were synthesized and tested against A2780 and A549 [41]. Cytotoxicity was measured versus healthy MRC-5 fibroblasts as well. Both mono- and di-functionalized complexes manifested significant photocytotoxicity after irradiation with blue light with IC_50_ values in the nanomolar range (IC_50_ = 0.11–7.1 for A2780 and 1.2–51.9 for A549). At equimolar concentrations, the di-functionalized complexes caused higher platinum cell accumulation and photogenerated ROS, compared to their mono-functionalized analogs. Another diazido-Pt(IV) complex [42] (Figure 10) was unreactive in the dark, but highly cytotoxic when irradiated with visible or UV light (MTT test, 2 h exposure). It acted as a prodrug that, upon photoactivation, was reduced to square planar Pt(II) species that binds to nuclear DNA. Additionally, photoactivation causes the release of a variety of reactive species such as azidyl and hydroxyl radicals, singlet oxygen and others that further add to the cytotoxic effect and possibly cause immunogenic cell death.

When photoactivated, the complex is able to induce calreticulin exposure on the membrane of A2780 cells as well as the release of high mobility group box 1 (HMGB1) protein and ATP to the extracellular environment—known symptoms of immunogenic cell death.

Essential information on the Pt complexes’ structure, type of cancer cells suppressed and biological activity has been summarized in Table 1.

### 2.2. Ruthenium Coordination Compounds

Platinum-based drugs are undeniably effective in the treatment of a variety of cancers. Unfortunately, they are not a “silver bullet”—limited activity against many common neoplastic diseases, significant toxicity and acquired platinum resistance [43] have all pushed the broader scientific community to look for alternatives. One such alternative is presented by another member of the transition metal family—ruthenium. It is redox-active, i.e., exists in a variety of oxidation states, the most prominent under physiological conditions being +2 and +3 (the former is considered more active) [44]. Ru(II)/(III) complexes are characterized by six-coordinated octahedral configuration, allowing for employment of biologically active ligands with different geometries. Rates of ligand exchange in Ru(II) and Ru(III) complexes are similar [45] to those of Pt(II). Ru compounds are considered less toxic than Pt. Ionic mimicry (similar ionic charge/ionic radius ratios) allows Ru to compete with Fe for binding with biomolecules, employing the transferrin pathway to specifically target cancer cells [46]. Protein and DNA binding, impairment of mitochondrial functions, cell cycle arrest and apoptosis are frequently described. Photoactivation is also a prominent feature of Ru-complex research. It provides improved toxicity over a localized area, thus reducing systemic adverse events while promoting antiproliferative activity. Two main approaches toward synthesis of Ru-based anticancer complexes seem to prevail above all. The first one involves complexation with polypyridyl-type bi/tri-dentate ligands, employing a bipyridine, terpyridine or 1,10-phenanthroline scaffold. The second approach involves coordination with a five or six-membered arene ligand, a monodentate and a bidentate ligand. To the reader’s benefit, the authors recommend a number of detailed reviews, delving deep into the subject of ruthenium complexes in anticancer therapy [43,44,47,48]. Henceforth the authors introduce some prominent samples of the vast experimental and publication efforts that have been invested over the past three years into the search for Ru-coordination compounds with anticancer activity.

A Ru(II) complex with a Schiff base and p-cymene as ligands was synthesized and tested against Caco-2 and normal mouse fibroblast L-929 strains [49]. The Schiff ligand itself and its ruthenium complex manifested low antiproliferative activity against Caco-2, the former being the less toxic of the two (IC_50_ = 803.65 and 510.26, respectively).

Cole and coworkers [50] synthesized a series of Ru complexes, two 6,6′-dimethyl-2,2′-bipyridine and an imidazo[4,5-f][1,10]phenanthroline ligand as potential photodynamic therapeutic agents (Figure 11). Cytotoxicities in dark conditions as well as photocytotoxicities were estimated using the resazurin viability assay in both hypoxic and normoxic conditions. All compounds were moderately active in dark conditions against SKMEL-28 melanoma cells (EC_50_ between 30−75 μM). Hypoxic conditions caused EC_50_ values to double, consistent with previous observations.

Irradiation with visible light increased cytotoxicity of compounds 1, 4, 6 and 7, dropping EC_50_ to between 2.5 to 5 μM. Compounds 1 and 7 maintained their high post-irradiation cytotoxicity even in hypoxic conditions. The authors tried irradiation with red and green light, with less success. When the substitute R was replaced by a chain of three or four thiophene rings (complex 8), the complex exhibited attomolar (in normoxia) and picomolar (in hypoxia) cytotoxicities toward SKMEL-28 cells [51]. The authors proposed the photosensitizing capacity of these complexes to be a result of efficient production of singlet oxygen and a possible involvement of triplet intra-ligand charge transfer states. Similar polypyridine ruthenium(II) complexes [52] were tested (MTT assay) against melanoma B16, Hep-G2, A549 cancer and LO2 normal cell lines. The antiproliferative effect was in the low micromolar range. Particularly against B16, the compounds performed better than cisplatin both in terms of antiproliferative effect and in terms of selectivity. Cell migration was disrupted, G0/G1 cell cycle arrest was observed and apoptosis was increased. B16 cells also experienced a significant increase in intracellular ROS. The Ru(II) complexes penetrated the mitochondrial membrane. Intracellular GSH was depleted and MDA levels were significantly increased compared to the non-treated control group.

Another novel photosensitizing polypyridyl complex incorporates the anthraquinone rhein [53] (Figure 12). The complex is lipophilic and able to penetrate the lysosomes of A549 cells.

In dark conditions, the complex does not generate intracellular ROS. Light irradiation causes intracellular ROS production. Cytotoxicities against MCF-7, A549, leukemia NB-4, A2780, cisplatin-resistant A2780R and normal liver cells LO2 were measured using the MTT assay in both light and dark conditions. IC_50_ values against all cell lines are moderate (35 to 250 μM) after 48 h of cultivation in darkness. Furthermore, 15 min of irradiation increases toxicities dramatically (IC_50_ of 35 to 25 μM). Phototoxicity indices against all cell lines are between 4 and 28.5. The authors propose the main mechanism of action in light conditions is the induction of autophagy.

Similar polypyridyl Ru(II) complexes have been synthesized, bearing a naphthoquinone moiety (plumbagin) [54]. The two most potent complexes bear two 1,10-phenanthroline plus one plumbagin ligands and two 4,7-diphenyl-1,10-phenanthroline and one plumbagin ligands. Increasing the lipophilicity of the ancillary ligands (from dimethyl sulfoxide (DMSO), through bipyridine and 1,10-phenanthroline to 4,7-dipjenyl-1,10-phenanthroline) increases biological activity against in vivo MCG-803 tumor mice model. The two most potent substances severely impair mitochondrial respiration and glycolysis, induce DNA damage and increase expression of the growth arrest and DNA damage inducible alpha (GADD45A) gene, causing G0/G1 cell cycle arrest.

Notaro and coworkers investigated a polypyridyl Ru(II) complex, incorporating a maltol ligand with the aim to improve bioavailability of the metal ion [55]. The compound was highly cytotoxic against HeLa, A2780 and its cisplatin and doxorubicin-resistant varieties A2780cis and A2780ADR, CT26, CT26LUC cancer cells and normal RPE-1 cells, manifesting no selectivity toward cancer cells. Adding the maltol moiety improved cytotoxicity of the complex (IC_50_ = 0.42–2.86 μM), even though maltol itself is non-toxic. The compound induces apoptosis in HeLa cells after 4 h of incubation, cell accumulation being greater than that of cisplatin. It accumulates predominantly in the cellular nucleus. Another study [56] discovered that cellular resistance to ruthenium cyclometallated compounds depends on ABCB1 export and EGFR gene expression. Inhibiting these genes improved the biological activity.

Chen and coworkers tested a Ru(II) polypyridyl complex with intrinsic antiproliferative properties in combination with taxol [57] against HeLa and A549 cells and their taxol-resistant variants. Synergistic effects were observed, even at low dosages. Pyroptosis, caspase 1 activation, gasdermin D (GSDMD) activation and ROS increase in HeLa/Taxol cells was observed. An in vivo naked mice model also yielded promising results.

A series of N-heterocyclic carbene Ru(II) arene complexes [58] were synthesized and tested for antiproliferative activity (Figure 13).

After 48 h of exposure, results from the MTT assay demonstrated that attaching short alkyl moieties caused very low cytotoxicity. Complexes 4 and 6 had the highest lipophilicity and manifested high cytotoxicity, comparable to cisplatin, against a panel of cancer cell lines—A549, HT-29, HCT116, LoVo, HeLa and A2780 (IC_50_ = 1.98−25.6). They exhibited antimigratory and proapoptotic activity against A2780 and induced mitochondrial dysfunction, releasing intracellular ROS.

Morais and coworkers tested a series of cyclopentadienyl Ru(II) complexes [59] with monodentate imidazole-based or bidentate heteroaromatic ligands. The investigated substances were very lipophilic. Binding to human serum albumin (HSA) was noted. Those that incorporate a bidentate heteroaromatic ligand manifested significant antiproliferative activity (MTT assay, 72 h) against A2780 (IC_50_ = 0.20−0.45 μM), MDA-MB-231(IC_50_ = 13.4 to over 100 μM) and HT-29(IC_50_ = 11.3 to over 100 μM), unlike the group with monodentate ligands.

Elsayed and coworkers [60] synthesized one Ru(II) and one Ru(III) complex, coordinated with 2-aminophenyl benzimidazole and DMSO (Figure 14-1 and Figure 14-2, respectively)

In vitro MTT assay showed that both complexes manifested moderate cytotoxicity against Caco-2 and MCF-7 (IC_50_ = 230–320 μM) and very low toxicity against non-cancerous liver (THLE-2). Activity of the Ru(III) complex—complex 2—was higher than that of its Ru(II) analog. Treatment with complex 2 induced DNA laddering in both cancer cell types tested. G2/M cell cycle arrest was observed. An in vivo mouse model with Ehrlich ascites carcinoma (EAC) showed that complex 2 decreases liver damage markers in inoculated mice. The authors proposed that this complex prevented the growth of the EAC cells and caused cell death. Superoxide dismutase (SOD), catalase (CAT) and glutathione (GSH) levels increased in a dose-dependent manner compared to the non-treated group of animals.

A series of Biginelli hybrids were synthesized and liganded to Ru(II) [61]. The antiproliferative activity (MTT, 48 h exposure) against a panel of cancer cell lines of the series of ligands was moderate. Adding Ru(II) significantly increased potency. Two of the complexes (Figure 15) manifested significant cytotoxicity against HeLa, A375 and K562 cancer cells (IC_50_ = 8.63−33.85 μM).

Adding chlorine or nitro-groups to the aromatic ring seemed to increase cytotoxicity. These two complexes demonstrated pronounced antiangiogenic effects, via in vitro inhibition of endothelial cell tube formation after 48 h. Cell migration was also impeded.

A series of Ru(II)-p-cymene-imidazophenanthroline complexes [62] were synthesized and tested against the HeLa and CaCo-2 cancer cell lines and normal HEK-293 cells. CT-DNA and bovine serum albumin (BSA) binding was noted. The most prominent member (Figure 16) exhibited strong antiproliferative activity (MTT assay) at an IC_50_ of 2.0−2.5 μM. Toxicity against HEK293 was found to be low with selectivity factor of over 40.

Strong antiproliferative activity was observed even in the presence of 1 mM GSH. Loss of activity was minimal. Adding electron-donating groups to the hydroxyphenyl moiety decreased biological activity, while addition of electron-withdrawing groups increased cytotoxicity. Similar complexes, incorporating substituted phenyl, instead of a substituted hydroxyphenyl moiety [63] were tested against the MDB-MA-231, HeLa and normal HEK293 cell lines. CT-DNA and BSA binding was noted here as well. The antiproliferative activity of the two most prominent substances toward cancer cells was similar to cisplatin (MTT assay); however, their toxicity toward the normal cell line was significantly lower—IC_50_ values of 85 and 178 against cisplatin’s 64. A p-fluorophenyl moiety improved activity against HeLa and p-nitrophenyl moiety increased toxicity against MDB-MA-231.

Ru(II) complexes with diclofenac and organophosphines as ligands [64] (Figure 17) were synthesized and tested for biological activity. MTT testing revealed the compounds to be effective against A549, MDA-MB-231 and MCF-7 cancer cells, with an IC_50_ from 0.56 to 12.28 μM—similar to or lower than cisplatin.

The tested compounds were more toxic to the non-cancerous MRC-5 and MCF-10A cell lines compared to cisplatin. Interactions with BSA and CT-DNA were noted. One of the complexes induced apoptosis in MCF-7.

Two bis-aminophosphine Ru(II) complexes with p-cymene ligands [65] exhibited antiproliferative effects against A375 cells with IC_50_ values (6.72 and 8.76 μM) lower than cisplatin. Both compounds were more active than the bis-aminophosphine ligand alone. The complex, containing two units of p-cymene-Ru(II) induced apoptosis in the tested cell line.

A number of Ru(II) p-cymene complexes with cyclic/polycyclic aromatic diamine ligands [66] manifested better cytotoxic effects than cisplatin against the OVCAR-3, M-14 and HOP-62 cancer cell lines (IC_50_ = 4.31–6.31 μM). CT-DNA binding improved with the increase in the delocalization of the aromatic fragment of the ligand.

A series of dinuclear p-cymene-tetrazole-Ru(II) complexes [67] were tested against several types of cancer cell lines (MTT assay). The ligands themselves were non-cytotoxic. Two of the compounds yielded promising results against the HeLa, MCF-7 and A549 cell lines. DNA binding, bovine serum albumin binding and morphological changes indicating apoptosis were observed. G0/G1 cell cycle arrest was induced and cell migration was inhibited.

A number of half-sandwich arene chloride complexes of Ru(II) [68] were bound to CT-DNA and BSA protein. All complexes exhibited good anticancer potency against MCF-7 cells (IC_50_ = 2.64–18.21 μM).

Organometallic Ru(II)-flavone [69] complexes were synthesized and tested against the MCF-7 and MDA-MB-231 cell lines. Neither complex exhibited significant antiproliferative activity. Interestingly, one of the thioflavone ligands, containing a C=S group demonstrated very high cytotoxic activity against MCF-7. Once this functional group was engaged in complexation with Ru(III), anticancer activity was diminished, though still significant (IC_50_ = 1.2–43.06 μM). The Ru(II)-thioflavone complex was found to inhibit MCF-7 and MDA-MB-231 cell migration.

Ru(III)-pyrazolopyrimidine [70] complexes manifested stronger antiproliferative activity against SCOV-3 cells compared to cisplatin. Toxicity against normal liver cells was low. Cell invasion and proliferation were inhibited. Intracellular ROS levels increased, mitochondrial membrane potential was reduced and apoptosis was initiated.

Ru(II) complexes with biologically active aminoflavone ligands [71] were synthesized in order to overcome cisplatin resistance in a number of cell lines. The compounds (Figure 18) were tested against A2780, A2780cis, Toledo and Toledo-cis.

They were effective in the low-micromolar range both against sensitive and cisplatin-resistant lines (MTT assay, IC_50_ = 0.5–4.6 μM). Mitochondrial membrane potential loss occurred in a dose-dependent manner. Apoptosis was induced due to interaction with DNA.

Conjugates containing two or three dinuclear Ru(II)–arene structures [72] have been tested against the A2780, A2780cisR (cisplatin-resistant), A24 and (D-)A24cisPt8.0 cell lines. All compounds are much more potent than cisplatin with an IC_50_ between 23 and 650 nM. Ester conjugates are more potent than amide analogs.

Qu and coworkers investigated how protonation states of ligands in Ru(II) complexes may influence photodissociation and quantum yields for singlet oxygen [73]. They observed that deprotonation of bipyridine ligand hydroxyl groups increased quantum yield for singlet oxygen (these are reduced ten-fold by protonation) and decreased quantum yield for photodissociation products.

Oliveira et al. prepared two Ru(II)-diphosphine complexes, containing lapachol and lawsone as ligands [74]. Both complexes were highly cytotoxic against the MDA-MB-231, MCF-7, A549 and DU-145 (prostate) cancer cell lines with IC_50_ values in the low micromolar range (IC_50_ = 0.03–2.70 μM). They interacted with the minor DNA grooves and moderately with bovine serum albumin. The lapachol-containing complex was highly selective against triple-negative breast cancer (MDA-MB-231), inhibiting cell migration and colony formation. It arrested cell cycle, disrupted mitochondrial membrane potential and caused an increase in ROS.

Ru(III) coordination compounds with quinolone antibiotics [75] were tested against LoVo colon cancer in order to translate antibacterial action to anticancer activity. Complexes with levofloxacin, ciprofloxacin and ofloxacin demonstrated stronger antiproliferative effect than cisplatin. G0/G1 cell cycle arrest was observed. More Ru(III) complexes with triazolopyrimidine [76] ligands were screened on MCF-7, HeLa and normal L929 cells. Increased ROS generation was observed as well as DNA and protein binding. Increased lipophilicity of the tested substances was associated with higher antiproliferative effect (IC_50_ as low as 4 μM against MCF-7 and 5 μM against HeLa). Toxicity against L929 was lower than cisplatin’s.

A Ru(III) complex with 1,4,8,11-tetraazacyclotetradecane exhibited an antiproliferative effect against SiHa cells, with an IC_50_ of 48 μM after 48 h of exposure [77]. Cell morphology changes were observed, such as nuclear fragmentation. The number of apoptotic cells increased, while the number of viable cells decreased after 24 h of exposure at 50 μM concentration.

Essential information on the Ru complexes’ structure, type of cancer cells suppressed and biological activity has been summarized in Table 2.

### 2.3. Gallium Coordination Compounds

Gallium is a post-transition metal that exhibits a typical oxidation state of +3. In terms of ionic radius, electric charge and coordination number, Ga(III) closely resembles Fe(III) [78]. For that reason, similar to Ru, Ga compounds are able to compete for iron-occupied sites in biomolecules. This ionic mimicry is fundamental for the biological activity of Ga(III) substances. The involvement of gallium in cancer research dates back four decades [79,80,81]. Gallium compounds have been applied as diagnostic and therapeutic agents in oncology [79]. Rapidly proliferating malignancies have high metabolic activity and need significant iron intake. Many types of cancer are characterized with overexpression of transferrin receptors [82]. Ga(III) competes with iron for transferrin binding which allows it to penetrate “iron-hungry” cancer cells and exhibit its physiological effects—impairment of DNA synthesis, disruption of mitochondrial function, overall inhibition of iron-dependent enzymes, generation of ROS and ultimately, apoptosis [83]. As the search for novel, metal-based anticancer drugs intensifies, scientific interest into Ga(III) complexes with antiproliferative action has been revitalized over the past decade [83,84,85,86,87,88,89]. The relatively low number of studies, performed so far, combined with the intrinsic anticancer properties of the Ga(III) ion itself present an exciting possibility for research into novel complexes with biologically active ligands and potential antiproliferative action.

A number of gallium complexes with planar tetradentate ligands have recently been synthesized. Unlike the most prevalent six-coordinated complexes, these tend to block interactions between the metal coordination center and biomolecules to a lesser extent. In this case the vacant coordination sites may possess labile solvent molecules that would, in theory, allow for improved interaction with biomolecules by way of solvent ligand exchange. A number of such gallium(III) salens were synthesized by Zhang and co-workers [9]. One of the complexes (Figure 19) exhibited anticancer activity against several cell lines (HeLa, Hep-G2, MCF-7, A549) within the nanomolar dosage range (IC_50_ = 0.42–1.25 μM), as established using MTT assay.

Coordination with gallium(III) dramatically improved activity compared to free ligands and zinc analogs. The ethyl substituent at the amino-group turned out to be crucial, methyl and 1,4-butanediyl moieties decreased biological activity. The most prominent complex was found to enter cells through passive diffusion, being distributed mainly in mitochondria, endoplasmic reticulum and lysosomes. It was also discovered to inhibit protein disulfide isomerase in a concentration-dependent manner.

Gross and coworkers developed tetradentate metallocorroles with potential anticancer activity [90]. A variety of metal ions were coordinated with a corrole ligand. All compounds manifested moderate cytotoxicity against the DU-145, SKMEL-28, MDA-MB-231 and OVCAR-3 cell lines. Cytotoxicity increased with lipophilicity which depended on the type of liganded metal ion. The Ga(III) complex exhibited moderate antiproliferative activity (IC_50_ = 129–274 μM).

A number of benzoylpyridine thiosemicarbazones were synthesized as tridentate ligands with potential antiproliferative properties [91]. After synthesis, these series of ligands were coordinated with gallium(III) in a molar ratio of 1:1. Both the ligands themselves and their complexes exhibited significant antiproliferative activity (higher than that of cisplatin) against the HepG-2 cell line. Increasing the lipophilicity of the ligands tended to improve the antiproliferative effect after 48 h of treatment. Additionally, the inclusion of gallium(III) further improved the observed action. The most potent ligand and its gallium(III) complex (Figure 20) significantly increased early and late apoptosis in that same cell line at 5μM concentrations. Mitochondrial membrane potential was decreased. Ferritin expression was downregulated, while transferrin receptor-1 expression was upregulated. Both activated caspase-3 and ROS were increased, the effect of the gallium(III) complex being significantly improved, compared to the ligand itself.

Firmino and co-workers [92] synthesized two isonicotynoyl hydrazones as potential iron chelators. Generally, iron chelators tend to exert antiproliferative effects as they deprive malignant cells of much necessary iron. Both compounds (Figure 21) were coordinated to gallium(III) at a ligand: metal ion ratio of 1:2.

The novel ligands and their gallium(III) complexes were assayed for antiproliferative action against the leukemia HL60, MCF-7, HCT116 and PC-3 cancer cell lines as well as the non-malignant HEK-293 line (MTT, 48 h). The hydrazone, containing primary amino-group (compound 2) and its gallium(III) complex did not inhibit cell viability at 10 μM concentration. When that amino-group was replaced with a methyl moiety (compound 1), cell viability in all cancer cell lines was reduced to 50% at that concentration. Clonogenicity was reduced. Increasing lipophilicity improved anticancer activity. It is noteworthy that the activity of both ligands did not differ significantly from the activity of their respective complexes. Compound 1 and its complex were further tested for cytotoxicity. Their IC_50_ values against HL60 and HCT116 were within the low micromolar range (IC_50_ = 0.4–2.0 μM), exhibiting at least 25-fold lesser toxicity toward the non-malignant HEK-293 line.

Two similar hydrazones were complexed with gallium(III) (Figure 22) [84] in order to test their anticancer and antitubercular activities. Both ligands and their respective complexes were tested for their impact on cell viability of cancer (MCF-7, PC-3) and non-cancerous (RWPE-1) cells with the help of the MTT test. All substances manifested cytotoxicity within the micromolar range against MCF-7, with gallium(III) complexation strengthening the observed effect. Ligands and complex 1 were inactive toward PC-3, while complex 2 has a selectivity index (related to RWPE-1) lower than 1. The improved activity of the halogenated ligand was associated with its higher lipophilicity. Its complex accumulated to a larger extent in both cancer cell strains.

A number of octahedral gallium complexes with polypyridyl ligands (Figure 23) were tested against bulk osteosarcoma cells (OSC) and osteosarcoma stem cells (OSCs) [93]. These compounds were found to be potent at nanomolar concentrations. Anticancer potency was measured via the MTT assay (IC_50_ = 0.07–3.60 μM). Salinomycin, cisplatin and carboplatin were used as positive controls. IC_50_ values of all three complexes were significantly lower than those of the positive controls. Activity improved with the increase of lipophilicity, complex 3 being active within the nanomolar range against osteosarcoma cells and osteosarcoma stem cells.

Assays were conducted also on non-cancerous cell lines—HEK 293T, MCF710A, BEAS-2B and GMO7575. Complex 3 was significantly less potent against these. It manifested an ability to enter osteosarcoma cell nuclei with a potential to damage genomic DNA and to initiate caspase-dependent cell death.

A series of publications [85,94] describe promising antiproliferative activity of a gallium(III) complexes with substituted 8-quinolinols. They are structural analogs of tris(8-quinolinolato)gallium(III), also known as KP46 and AP-002 (Figure 24a)—a potential metallodrug, currently undergoing clinical trials [94]. Gallium complexes of four structural analogs of KP46 (Figure 24b) were tested against several cancer cell lines—A2780, MDA-MB-231 and HCT116. The non-cancerous MRC5pd30 line was also investigated.

Complex 1 (with 5-chloro-substituted ligand) manifested the most pronounced antiproliferative effect (IC_50_ = 6.5–14.0 μM), comparable to that of the positive control cisplatin with regard to the various cell lines. Non-cancerous cells were affected at 25–65 times higher concentrations than cancer cells. The complex was significantly more active than the ligand itself.

A different study [95] focused on KP46 (Figure 24a) itself. It was tested against a variety of human cell lines: BJAB mock, including vincristine and doxorubicin-resistant; Nalm-6, including vincristine, etoposide and methotrexate-resistant strains; SK-N-AS and its vincristine-resistant subline; and K652. KP46 was found to induce apoptosis at a higher percentage than doxorubicin after 72 h of incubation. The observed activity was partially caspase-dependent, inducing processing of procaspases 3 and 9. The compound manifested a significant antiproliferative effect against BJAB and its vincristine-resistant subline, while the doxorubicin-resistant cells were unaffected. Mitochondrial membrane potential was decreased by about 64% at a 20 μM concentration. Increased pro-apoptotic Harakiri protein expression was observed. The compound was found to possess not only intrinsic antiproliferative properties, but also to be highly effective in combination treatments with “established” anticancer agents against resistant cell lines. Synergies with cytarabine and methotrexate, causing up to a four-fold increase of apoptotic cells when treating BJAB were observed. Even a low concentration of 0.75 μM, in combination with daunorubibin, was able to overcome daunorubicin resistance in the K652 strain—increases in the number of apoptotic cells were three-fold.

Essential information on the Ga complexes’ structures, type of cancer cells suppressed and biological activity has been summarized in Table 3.

### 2.4. Gold Coordination Compounds

Gold has been used as a therapeutic since ancient times [96]. The earliest recorded use of gold as a medicine dates back 2500 years in China. Over the past two centuries in Europe, gold compounds have been used to treat “melancholy”, fevers, syphilis and tuberculosis [96]. Compared to platinum-based drugs, complexes with gold tend to be less toxic. Such compounds have historically been applied with success in the treatment of rheumatoid arthritis (Myocrisin^TM^, Solganol^TM^, Ridaura^TM^ (auranofin), etc.) and malaria [97]. Chrysotherapy (therapy with gold-containing compounds) research in the field of oncology has been ongoing for several decades now as a possible alternative to platinum-based treatments and their numerous disadvantages in terms of severe side effects and cancer cell resistance [96,98]. Gold complexes as antitumor agents tend to target thioredoxin reductase (TrxR), and in general, proteins and enzymes bearing thiol groups [99]. Similar to ruthenium and gallium, research of novel gold coordination compounds with antiproliferative properties is driven by the distinct deficiencies of modern platinum-based therapies. Alkynyl-gold complexes are currently a subject of particular interest due to the particular stability of the C-Au bond, making the gold ion more resistant to physiological reductants before reaching its target. Additionally, modifying the ancillary ligand can impact the pharmacological behavior of the investigated complex [98]. The authors would like to recommend the following detailed reviews of Au(I)/(III)-complex applications in oncology [99,100,101]. The most recent discoveries over the past three years are described below.

A number of halo and pseudohalo gold(I) complexes with 4,5-diarylimidazoles (Figure 25) were synthesized and tested for antiproliferative activity against hepatocellular carcinoma strains HepG2, SMMC-7721 and Hep3B [102]. The study was an expansion on previous work with similar compounds [103].

MTT assay was applied to measure the antiproliferative effect with cisplatin and auranofin being used as positive controls. All investigated substances manifested antiproliferative activity similar to that of the controls (IC_50_ values were up to 20μM). Complex 1 was the least active. Notably, the larger the halogen ligand (I > Br > Cl), the stronger the antiproliferative activity, a fact attributed by the authors to an increase in lipophilicity in the same order. Acetato-gold complexes were more active than cyanate-gold complexes. The most active substance 6 was found to be stable in PBS, including in the presence of high concentrations of GSH (Au prefers “soft” S-ligands). It exhibited better selectivity than the positive controls toward hepatocellular carcinoma HCC cells compared to normal LO2 and H8 cells. TrxR, a prominent target for gold, was inhibited, intracellular ROS production was increased and mitochondrial function was impaired.

Walther and coworkers [104] investigated two gold(I)-N-heterocyclic carbene complexes for anticancer activity (Figure 26).

IC_50_ values were within the nanomolar range (sulforhodamine B (SRB) assay) after 48 h of incubation against OVCAR3, NCI-H522, HT29, T-47D and PC-3 lines (GI_50_ = 0.26–0.79 μM). Growth of PC-3 derived xenograft tumors was inhibited, mammalian TrxR was suppressed and nuclear protein Ki67 was reduced—a marker for inhibition of cell proliferation.

Gulzar and coworkers [105] synthesized a series of NHC-gold(I)-thione complexes (Figure 27) and tested them against the HCT-15, A549 and MCF7 cancer cell lines (MTT assay, 24 h). Cisplatin was the positive standard.

IC_50_ values (74.26 to 102.17μM) were about 2–3 times higher than the those for cisplatin. The authors concluded that thione ligands do not contribute to the antiproliferative activity of such gold(I) complexes.

Alkynyl-gold complexes have been intensively studied in recent years for their antiproliferative effect [98]. They tend to exhibit significant anticancer effects, associated with TrxR inhibitory activity [106,107,108,109]. A number of investigations to that effect have been carried out recently.

A phenanthrene bridge was substituted with two terminal alkynyl groups in order to build potential anticancer complexes with two gold(I)-phosphine centers [110] (Figure 28). The potential antiproliferative activity of both gold complexes was assessed against the MCF-7, HEPG-2, PC-3 and MOLT-4 cancer cell lines (SRB, 72 h). The phenanthrene-ethynyl “bridging” ligand had very low activity against all tested strains. Both gold complexes behaved in a similar manner to the positive control cisplatin (IC_50_ varied at 18–28 μM with all cancer cell lines). DNA binding was noted for both complexes and the ligand. The authors of the study proposed that substituting the phenanthrene skeleton at positions 9 and 10 would allow for modification of lipophilicity of the complexes improving their solubility—a common issue with alkynyl-gold compounds [111].

Another study [112] evaluated a number of alkynyl-gold(I)-triphenylphosphine complexes against cancerous HT29, IGROV1, HL60 and non-malignant I407 cells. The most active substance (Figure 29) exhibited antiproliferative activity in the low-micromolar range (IC_50_ = 3.3–7.9 μM), similar to that of the positive standard auranofin. It, however, turned out to be most toxic against the normal cell line (IC_50_ = 1.7 μM). Two other noteworthy compounds inhibited proliferation in IGROV1 and HL60 cells, while being non-toxic to I407. Both were binuclear complexes and their thioredoxine reductase inhibitory activity was significantly stronger than in the rest of the experimental substances that were mononuclear.

Marmol and coworkers synthesized and tested alkynyl-gold(I)-substituted 3-hydroxyflavones against a series of cancer cell lines [113]. Several alkyne-substituted ligands were coordinated with Au(I), together with either triphenylphosphane (PPh_3_), or 1,3,5-triaza-7-phosphaadamantane (PTA) (Figure 30). Their antiproliferative activity, expressed as IC_50_ against undifferentiated Caco-2/TC7 cells is within the micromolar range (IC_50_ = 1.5–7.68 μM, MTT assay).

Lipophilicity was higher in the PPh_3_ compounds, compared to the PTA-containing complexes. MTT and SRB assays were both utilized to measure cytotoxicity of all eight gold complexes against MCF-7, HepG5 and Caco-2/TC7 cells. Selectivity was assessed with the help of non-cancerous differentiated Caco-2 as a model of the intestinal barrier. Cisplatin and auranofin were used as positive controls. All complexes manifested antiproliferative effects against the cancer cell lines within the micromolar range (IC_50_ is higher than auranofin and lower than cisplatin). The more lipophilic series of PPh_3_-containing compounds suppressed proliferation in Caco-2/TC7 and HepG2 to a greater extent compared to the PTA series. This trend was reversed with MCF-7. Selectivity indices of the PTA complexes were similar to those of auranofin and cisplatin. Those of the PPh_3_-complexes were much improved. Complexes 1b and 2c were additionally examined. Complex 1b inhibited COX-2, thioredoxine reductase, glutathione reductase, increased ROS levels and triggered apoptosis after 24 h of incubation. Complex 2c inhibited COX-1, thioredoxine reductase, glutathione reductase, increased ROS levels and triggered apoptosis after 48 h.

A series of mononuclear phosphane-Au(I)-alkynyl complexes (Figure 31) were synthesized by Babgi and coworkers [114] with the aim to elucidate the impact of phenolic Schiff base addition on biological activity, including antiproliferative (SRB assay), HSA binding and thioredoxine reductase inhibition (in silico docking).

The aldehyde complexes (Figure 1-1,-2) suppressed the proliferation OVCAR-3 and HOP-62 cells (IC_50_ in the 12–16 μM range). Changing the aldehyde group with a phenolic Shiff base caused dramatic improvement of cytotoxicity (IC_50_ in the 5–9 μM range). The p-hydroxy substituted complex 4 was much more effective against OVCAR-3, compared to its o-hydroxy-substituted analogs. Within the scope of this investigation, substituting the triphenylphosphine with a tricyclohexyl moiety did not impact biological activity. Molecular docking calculations showed that substituting the aldehyde group for a Schiff base may change the binding site with human thioredoxin reductase. Changing triphenylphosphine with tricyclohexanephosphine increased HSA binding. Adding a phenolic moiety had the same effect.

An alkynyl-activated quinazoline carboxamide was synthesized in order to produce a series of alkynyl-Au(I) complexes [115] (Figure 32). Quinazoline carboxamides tend to bind to translocator protein 18kDa (TSPO), situated on the outer mitochondrial membrane [116]. They could be applied as chemo-/photo-sensitizers and diagnostic agents as TSPO is overexpressed in a variety of cancer types.

All three complexes were stable in PBS at 37 °C for over 72 h, including in culture media. Their cytotoxicities were tested with the help of the XTT assay. Bladder cancer lines 5637 and T24 were incubated with the complexes for 72 and 96 h. Compound 3 manifested modest activity against 5637 after 72 and 96 h. The IC_50_ values of complexes 1 and 2 were in the low micromolar to nanomolar ranges (IC_50_ = 0.17–12.40 μM). Complex 1 was more active than complex 2 with 5637 (72 and 96 h) and T24 (72 h). Complex 2 was more active with T24 after 96 h. Cellular uptake decreased in the following way 2 > 1 > 3. Complexes 1 and 2 activated caspases, in the case of the latter with delayed timing. Thioredoxine reductase inhibition increased in the following order 1 > 2 > 3.

Bian and coworkers [117] synthesized a series of alkyne-activated pentacyclic triterpene derivatives (betulinic acid, ursolic acid, clycyrrhetic acid and oleanolic acid). Triterpenes are known for their anticancer activity [118], attacking a multitude of targets.

The authors tested whether conjugating such substances with gold(I) and PPh_3_ would yield complexes with varied mechanisms of action beyond the expected thioredoxine reductase inhibition. All substances were tested against the MCF-7, HT-29, HepG3 and A2780 cancer cell lines (MTT assay) and were found to manifest moderate antiproliferative activity against most cell lines, except A2780 where the impact was significant (IC_50_ for A2780 was mostly between 25 and 40 μM). The oleanolic acid derivative (Figure 33) was found to have an IC_50_ against A2780 in the low micromolar range (10 μM)—similar to the positive controls cisplatin and auranofin. It inhibited thioredoxine reductase (both purified enzyme and cellular-A2780) to a lesser extent than auranofin, impaired mitochondrial function, increased cellular ROS production and induced endoplasmic reticulum stress.

Romanova and coworkers synthesized a series of gold(I) complexes with alkynyl-activated ibuprofen [119] (Figure 34) and tested them (MTT assay, 72 h) for antiproliferative activity against MCF-7, MDA-MB-231 and HT-29 cancer lines as well as MCF-10A non-cancerous cells.

The activity of both complexes was comparable to the positive controls cisplatin and auranofin, where the IC_50_ was in the low micromolar range (0.98–3.42 μM), and selectivity was very much improved with regard to the non-cancerous cell line. The N-heterocyclic carbenium (NHC) complex had lower antitumor activity than the triphenylphosphine one, but manifested much better selectivity in relation to the non-neoplastic MCF-10A cells. TrxR and glutathione reductase inhibition was observed as well as an increase in ROS.

Essential information on the Au complexes’ structures, type of cancer cells suppressed and biological activity has been summarized in Table 4.

### 2.5. Lanthanum Coordination Compounds

Lanthanum is the first member of the lanthanide series of *f*-transition metals. Its typical oxidation state is +3 and its coordination number can vary between 6 and 12. In terms of ionic potential La(III) resembles a multitude of “biological” ions (Fe(III), Ca(II), Zn(II), Mg(II)) and is therefore able to competitively replace them in ion-binding proteins [120]. A number of studies have related La(III) toxicity to impairment of zinc- and iron-dependent enzymatic systems, suppressing SOD, CAT and disrupting mitochondrial function [121,122,123]. Currently lanthanum is applied in medicine in the form of the phosphate binder lanthanum carbonate [124]. During the past decade, the popularity of La(III) complexes in cancer research has been steadily rising, mostly due to La’s ability to mimic biometals and to coordinate bioactive ligands [83]. As the search for La(III) complexes suitable for cancer therapy is slowly gaining traction [125], the authors would like to introduce the most prominent studies on this subject over the past three years.

Mohammed and coworkers synthesized a series of lanthanide complexes with two ferrocene-substituted Schiff bases [126]. Both La(III) complexes with these two ligands (1:1 metal ion:ligand molar ratio) manifested moderate antiproliferative activities (IC_50_ about 22 μM) against the MCF7 cell line (SRB, 48 h). Molecular docking studies suggested interaction with the 3HB5 breast cancer receptor.

A tetradentate Schiff base ligand (Figure 35) was coordinated with La(III) [127]. Both substances were tested for antiproliferative activity against MCF-7 and HepG2 (SRB, 48h exposure) and yielded promising results (IC_50_ = 23–50 μM). Adding La(III) to the Schiff base decreased anticancer activity against MCF-7 and dramatically improved it against Hep-G2.

A La(III)-5-fluorouracil complex (1:1 molar ratio) was synthesized and tested against the Caco-2 cell line (trypan blue exclusion assay). The presence of the La(III) coordination center significantly improved cytotoxicity, compared to 5-fluorouracil [128]. Molecular docking studies suggested interaction with site II of BSA.

A La(III) complex with tyrosine (metal: ligand molar ratio is 1:3) was tested against MCF-7 cells [129]. After 72 h of incubation (MTT assay), the complex manifested antiproliferative activity (IC_50_ = 21 μM) similar to the positive control cisplatin. Furthermore, the compound was found to be non-toxic to non-cancerous ADSC cells.

La(III) was complexed with N,N’-bis(2-aminoethyl)oxamide (Figure 36-1), 2,2′-bipyridine (Figure 36-2), 1,l0-phenanthroline (Figure 36-3) and dipyrido(3,2-a:2′,3′-c)phenazine (Figure 36-4) [130]. Four complexes were synthesized—[La(1)_2_(NO_3_)_2_](NO_3_), [La(1)(2)](NO_3_)_3_, [La(1)(3)](NO_3_)_3_ and [La(1)_2_(3)](NO_3_)_3_. All complexes manifested moderated cytotoxic activities (MTT assay against MCF-7), the highest activity being observed by [La(1)_2_(3)](NO_3_)_3_, with an IC_50_ = 22 μM. The bigger the aromatic planar structure of the ligand, the greater the cytotoxicity, attributed by the authors to DNA intercalation and cleaving.

A mixed ligand La(III) complex with 2,2′-bipyridyl and 5,7-dibromo-8-quinolinol manifested significant activity against SK-OV-3/DDP, NCI-H460, HeLa and HL-7702 cells [131]. IC_50_ is within the low micromolar range (MTT assay). The strongest cytotoxicity was against HeLa—about 2 μM.

Two La(III) complexes (Figure 37) with pyridine-2,6-dicarboxylate were tested against the HL60, HepG2, HT29 and normal HFF cell lines [132]. Oxaliplatin was used as a positive control. The complexes were more potent than the ligands themselves.

Complex 1 manifested a stronger antiproliferative effect compared to complex 2 against all cell lines (MTT assay). Complex 1 was particularly effective against HL60 (IC_50_ = 0.69 μM). Activity against Hep-G2 and HT-29 was moderate (IC_50_ = 20.76 and 90.34 μM, respectively). Both compounds were non-toxic against HFF. Complexes 1 and 2 increased ROS levels in HL 60 by 211% and 141%, respectively, the result being higher than that for the positive control oxaliplatin.

A La(III) complex with 2,2′-bipyridine [133] was tested against the MCF-7, A549 and the non-cancerous HFB cell lines (MTT test, 24 h). Molecular docking suggested interaction with site III in BSA. 5-FU and methotrexate were used as positive controls. The complex manifested a promising antiproliferative effect. IC_50_ was about ten-fold lower than the positive controls (3.25-3.99 versus 27.5–47.8 μM). Additionally, cytotoxicity against the healthy cell line was about two times lower than methotrexate and 5-FU.

Essential information on the La complexes’ structures, type of cancer cells suppressed and biological activity has been summarized in Table 5.

## 3. Discussion and Conclusions

Metal-based drugs are continually gaining ground in modern medicine, particularly in the field of oncology. Platinum, ruthenium, gold, lanthanum and gallium coordination compounds provide promising perspectives in the constant search for novel anticancer drugs. The research reviewed here helped the authors draw a number of conclusions:Depending on the element serving as coordination center, the complexes under review exhibit their antiproliferative activity via different pathways: DNA impairment via inter-/intra-strand crosslinks, mitochondrial function impairment, generation of ROS and apoptosis/necrosis. These effects result from a variety of biochemical and physicochemical mechanisms: DNA/protein binding, ionic mimicry, competitive inhibition of enzymes and photosensitization. This multitude of possible metal-dependent modes of anticancer action allows for consideration of potential combination therapies that improve effectiveness, avoid therapy resistance and reduce systemic toxicities;The activities of the metal coordination centers can be modified with a suitable choice of ligands. High cytotoxicity against tumors is not enough, unless it goes hand in hand with good selectivity and/or a suitable cancer cell targeting mechanism;Physicochemical mechanisms such as photoactivation, as in photodynamic therapy, allow for targeting specific areas of the body with the aid of a photosensitizer with low systemic toxicity. Developing transition metal complexes with suitable photophysical properties seems to be a suitable direction, both logical and necessary, in the search of novel anticancer treatments.

## Figures and Tables

**Figure 1 molecules-28-01959-f001:**
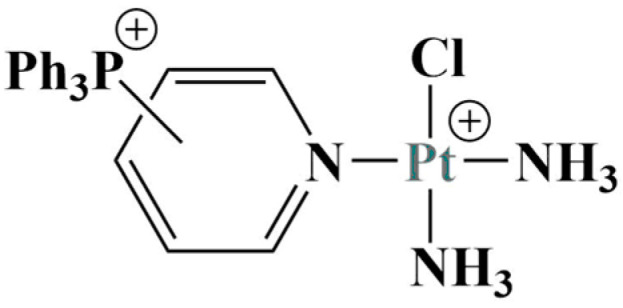
Modified pyriplatin, described in [22].

**Figure 2 molecules-28-01959-f002:**
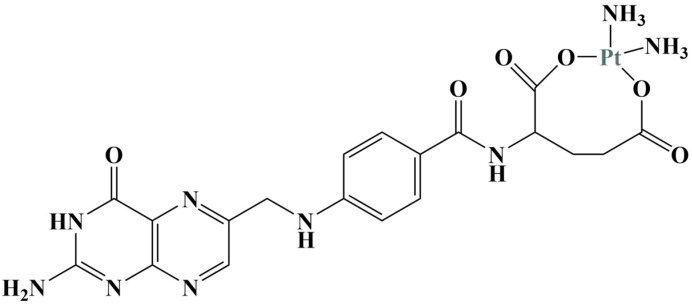
The folate-Pt(II) complex described in [24].

**Figure 3 molecules-28-01959-f003:**
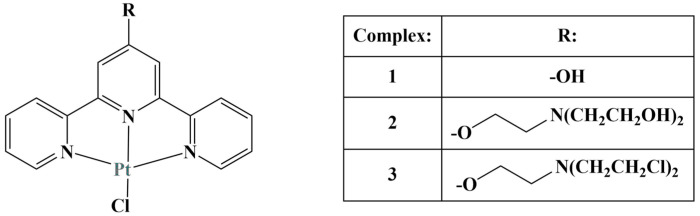
Three Pt(II)-terpyridine complexes, described in [25].

**Figure 4 molecules-28-01959-f004:**
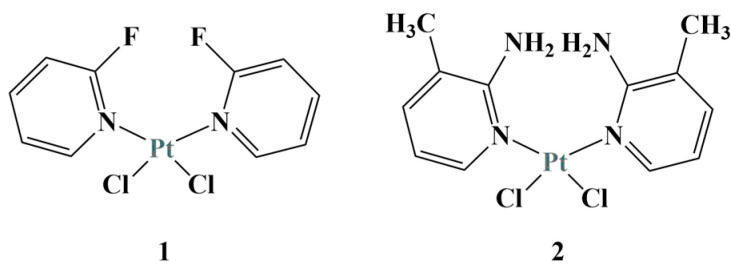
The pyridine-based complexes 1 and 2 described in [27].

**Figure 5 molecules-28-01959-f005:**
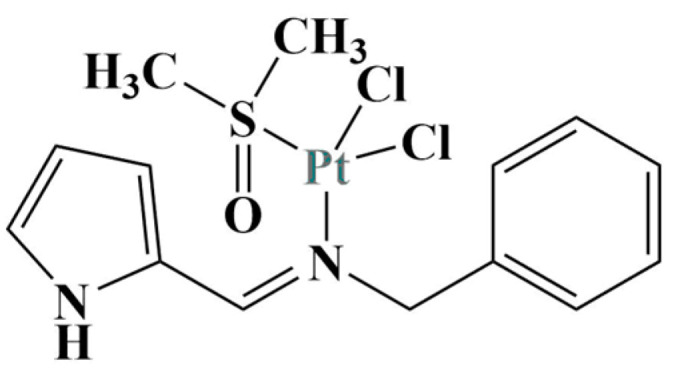
The most potent antiproliferative complex described in [28].

**Figure 6 molecules-28-01959-f006:**
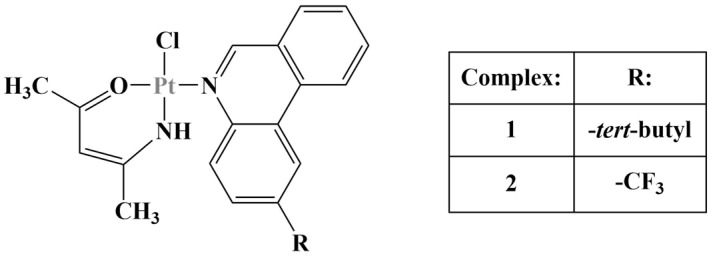
The complexes described in [30].

**Figure 7 molecules-28-01959-f007:**
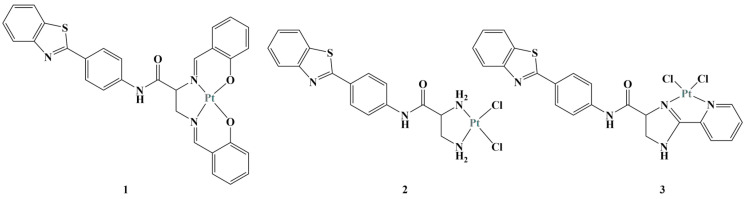
The complexes described in [32].

**Figure 8 molecules-28-01959-f008:**
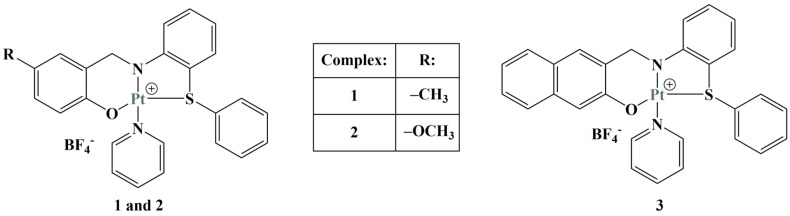
The complexes described in [33].

**Figure 9 molecules-28-01959-f009:**
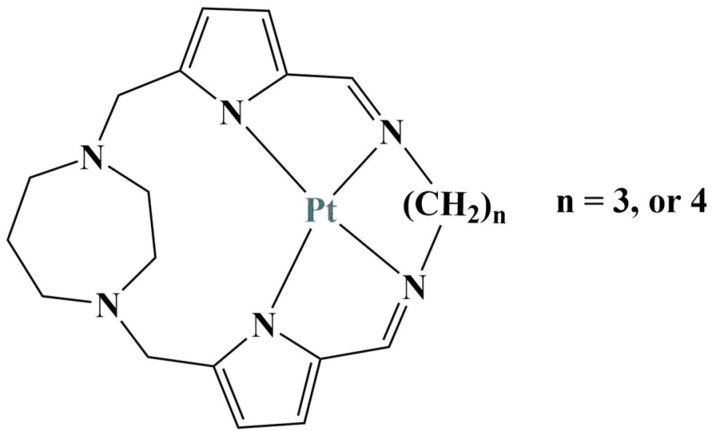
The macrocyclic Pt(II) complexes described in [36].

**Figure 10 molecules-28-01959-f010:**
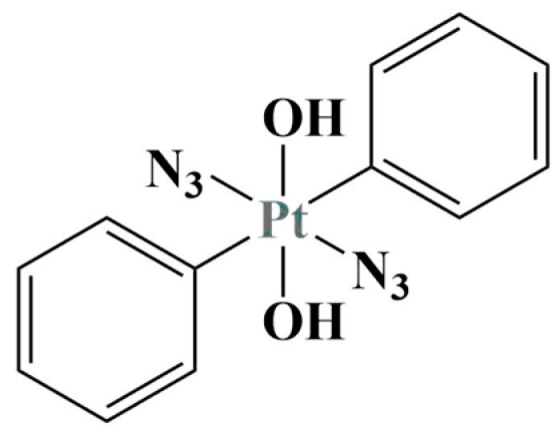
The diazido-complex described in [42].

**Figure 11 molecules-28-01959-f011:**
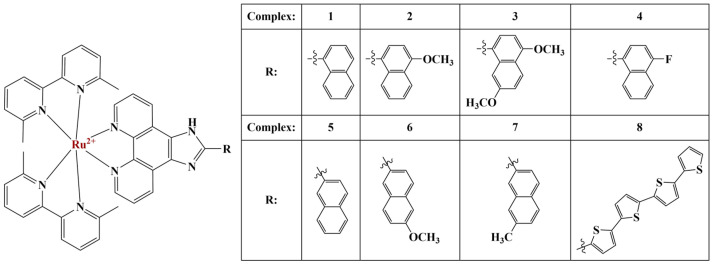
The complexes described by [50].

**Figure 12 molecules-28-01959-f012:**
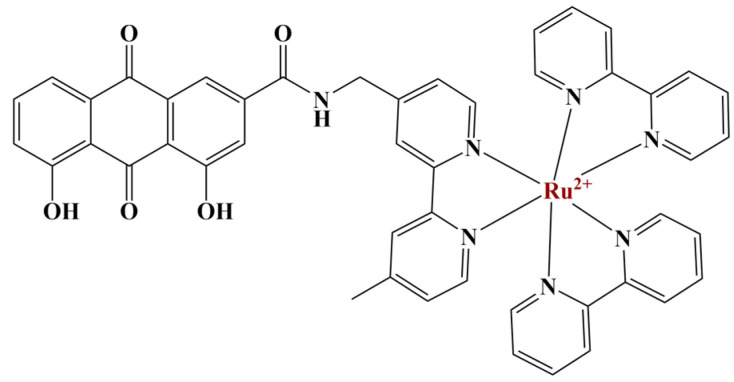
The complex described by [53].

**Figure 13 molecules-28-01959-f013:**
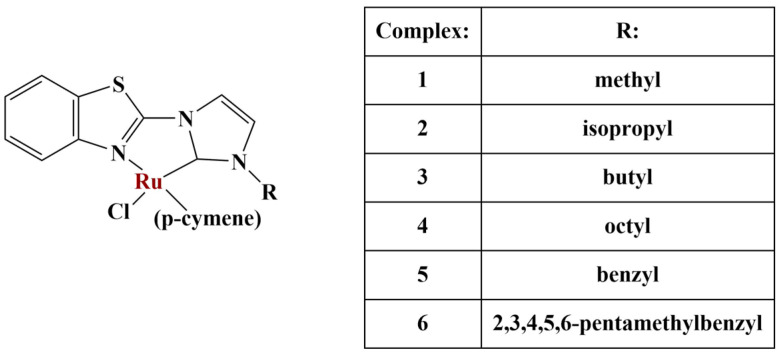
The complexes presented in [58].

**Figure 14 molecules-28-01959-f014:**
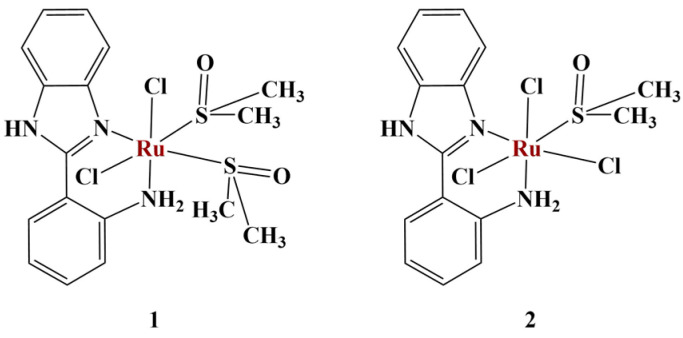
Complexes 1 and 2 presented in [60].

**Figure 15 molecules-28-01959-f015:**
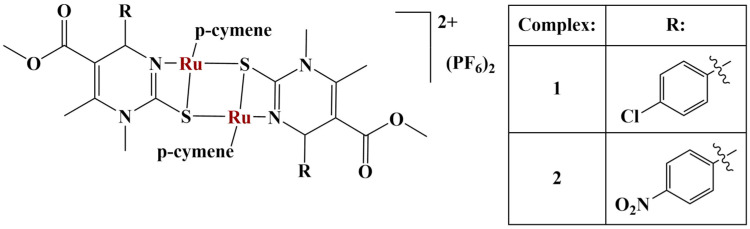
The complexes presented in [61].

**Figure 16 molecules-28-01959-f016:**
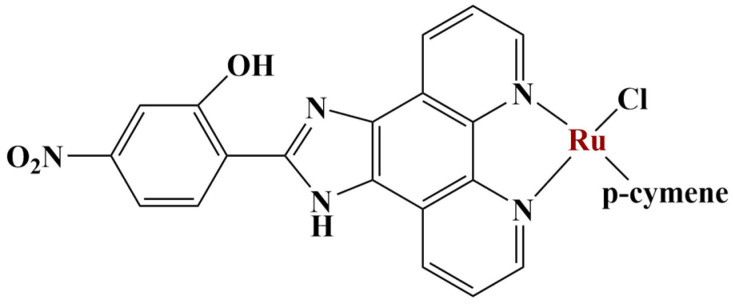
The most active complex presented in [62].

**Figure 17 molecules-28-01959-f017:**
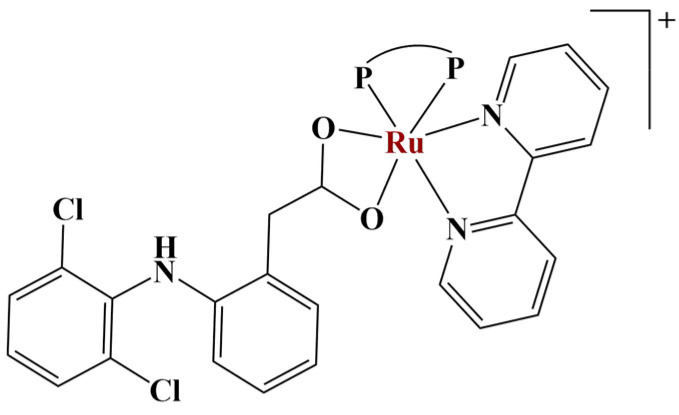
The complexes presented in [64].

**Figure 18 molecules-28-01959-f018:**
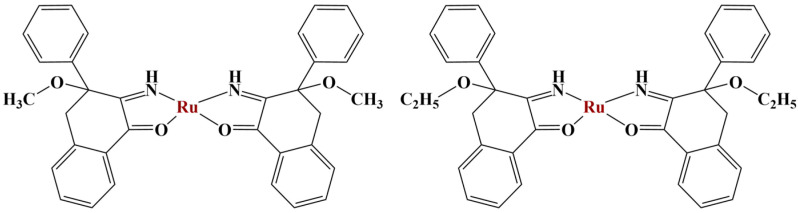
The complexes presented in [71].

**Figure 19 molecules-28-01959-f019:**
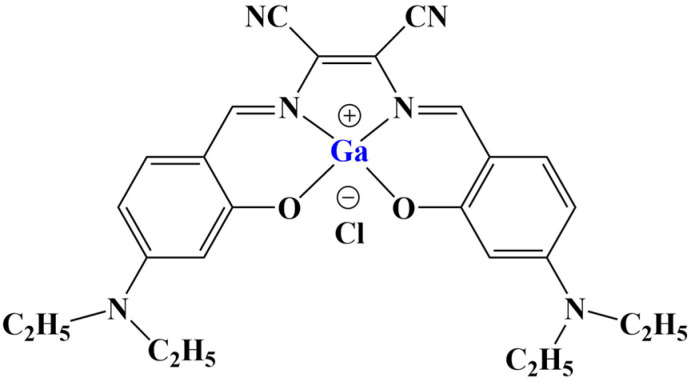
One of the complexes presented in [9].

**Figure 20 molecules-28-01959-f020:**
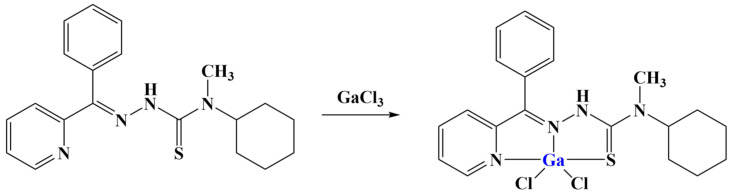
One of the complexes presented in [91].

**Figure 21 molecules-28-01959-f021:**
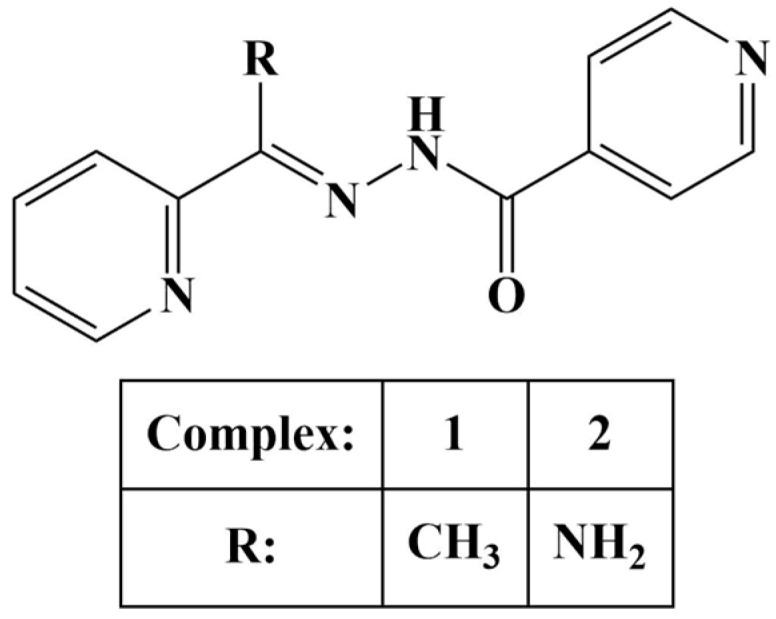
The ligands presented in [92].

**Figure 22 molecules-28-01959-f022:**
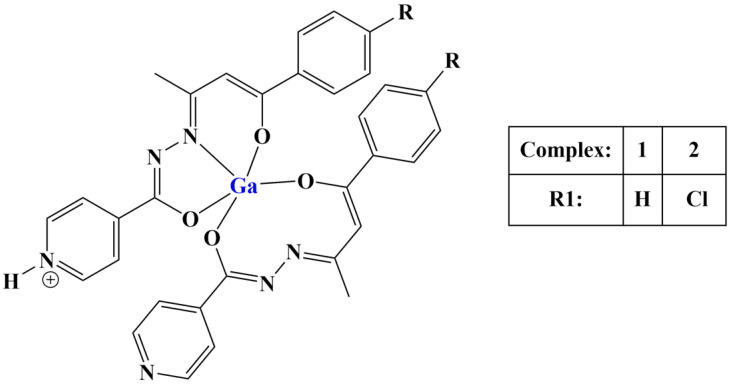
The complexes presented in [84].

**Figure 23 molecules-28-01959-f023:**
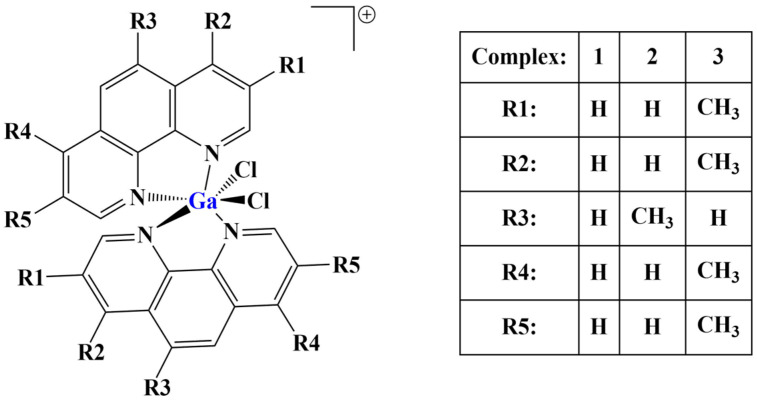
The complexes presented in [93].

**Figure 24 molecules-28-01959-f024:**
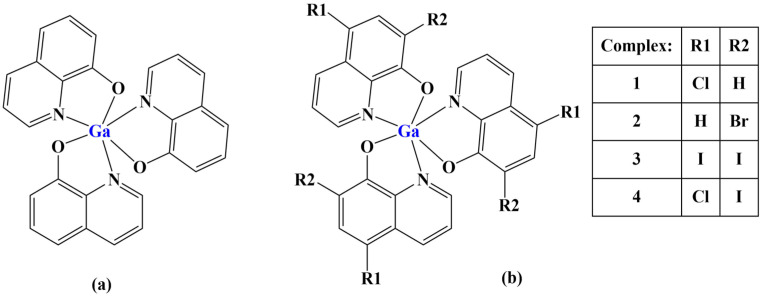
Structures of KP46 (**a**) and its analogues (**b**) presented in [94,95].

**Figure 25 molecules-28-01959-f025:**
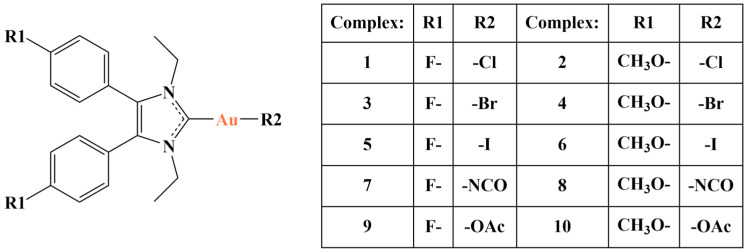
The complexes presented in [102].

**Figure 26 molecules-28-01959-f026:**
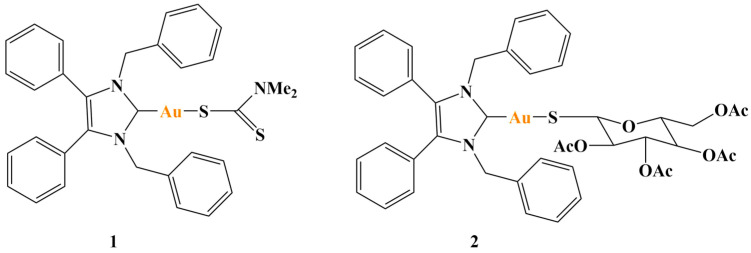
The complexes presented in [104].

**Figure 27 molecules-28-01959-f027:**
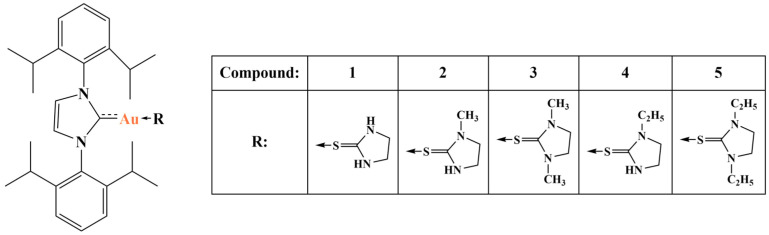
The complexes presented in [105].

**Figure 28 molecules-28-01959-f028:**
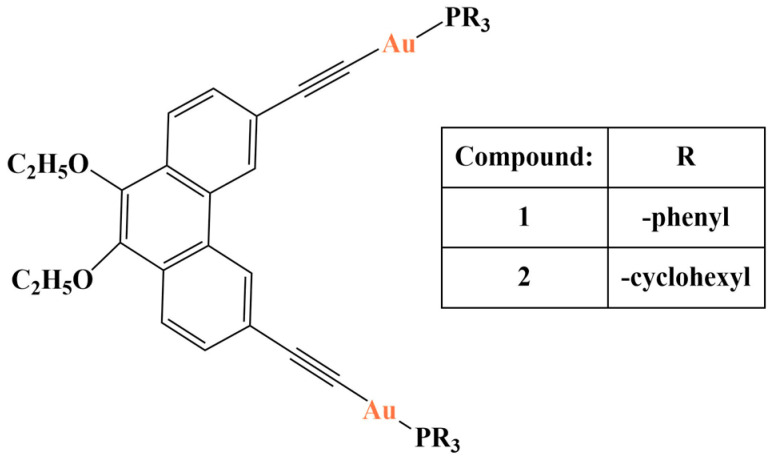
The complexes presented in [110].

**Figure 29 molecules-28-01959-f029:**
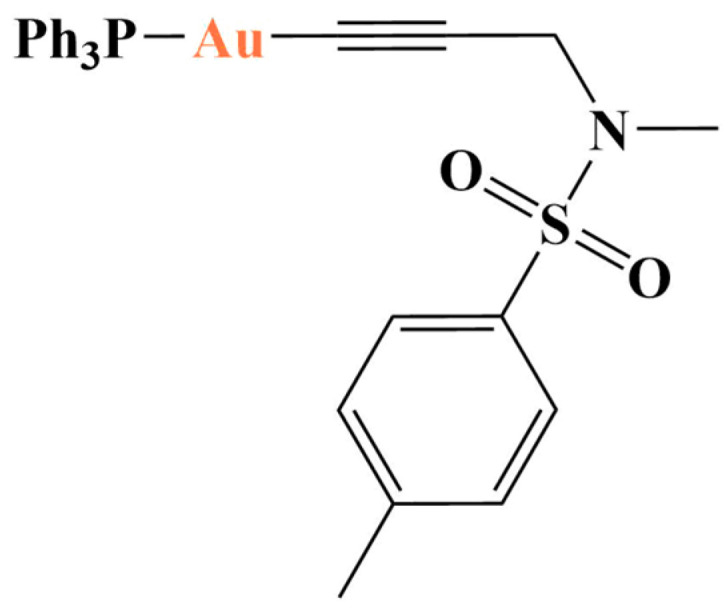
One of the complexes presented in [112].

**Figure 30 molecules-28-01959-f030:**
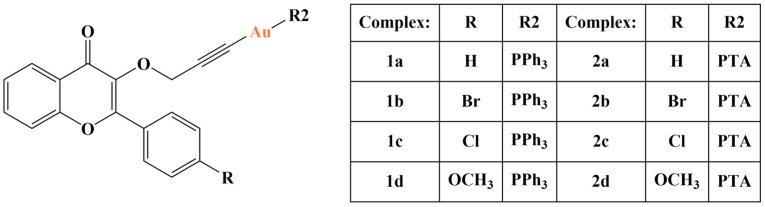
One of the complexes presented in [113].

**Figure 31 molecules-28-01959-f031:**
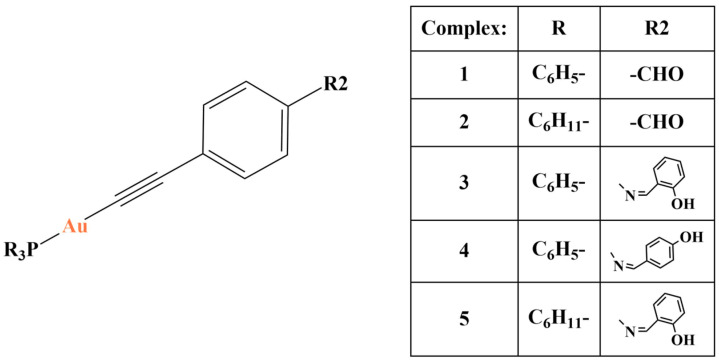
One of the complexes presented in [114].

**Figure 32 molecules-28-01959-f032:**
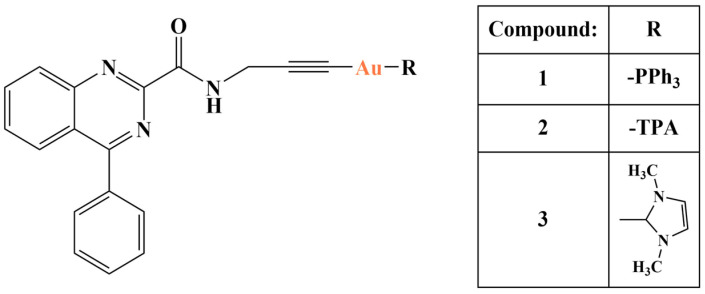
Some of the complexes presented in [115].

**Figure 33 molecules-28-01959-f033:**
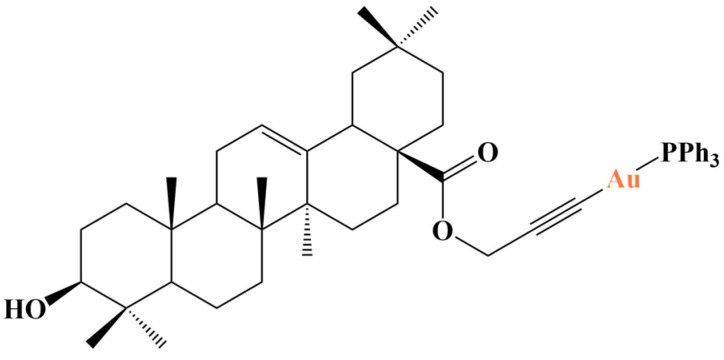
One of the complexes presented in [117].

**Figure 34 molecules-28-01959-f034:**
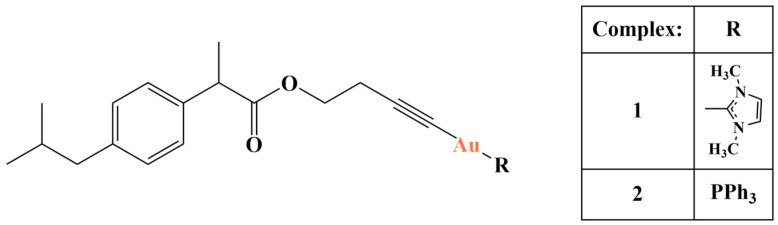
One of the complexes presented in [119].

**Figure 35 molecules-28-01959-f035:**
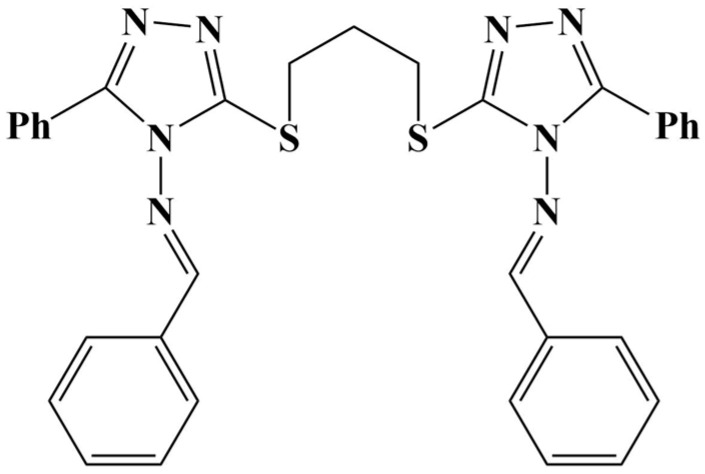
The ligand of the complex presented in [127].

**Figure 36 molecules-28-01959-f036:**
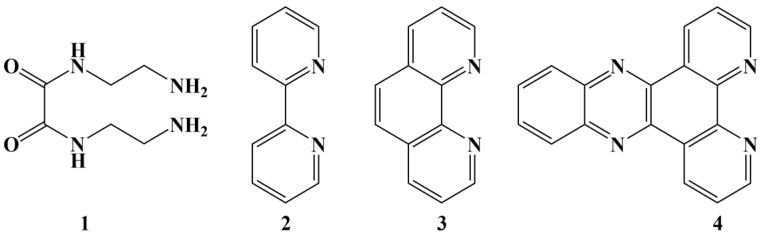
The ligands of the complexes presented in [130].

**Figure 37 molecules-28-01959-f037:**
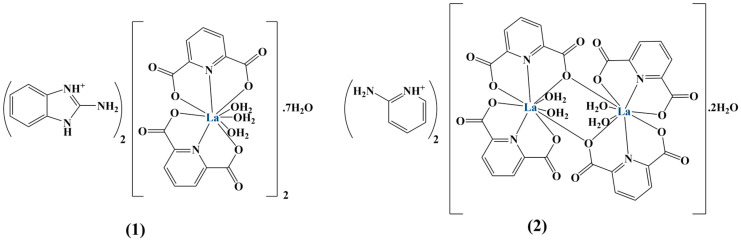
The ligands of the complexes presented in [132].

**Table 1 molecules-28-01959-t001:** Summary of the platinum complexes presented.

Citation Number	Metal Ion/Ligand Type	Effective against	Biological Activity (If Investigated)
[22]	Pt(II), triphenylphosphonium-pyridine	A549; HL-7702	mtDNA lesions, impairment of mitochondrial membrane potential
[23]	Pt(II), triangular polynuclear complex	HMLER; HMLER-shEcad; MCF10A	DNA intercalation; DNA groove binding
[24]	Pt(II), folic acid	MCF-7	increased Bak1/Bclx ratios and Caspase-3 activity
[25]	Pt(II), terpyridine	HCT116; SW480; NCI-H460; SiHa	Binding to L-histidine, 9-ethylguanine and L-cysteine
[26]	Pt(II), terpyridine	A549; A549/DDP; A431; HeLa; MCF-7	DNA binding; EGFR inhibition
[27]	Pt(II), substituted pyridines	DLD-1	n/a
[28]	Pt(II), pyrrole-substituted Schiff bases	Caco-2; HeLa; Hep-G2; MCF-7; PC-3	DNA intercalation
[29]	Pt(II), Schiff base	MCF-7	CT-DNA binding
[30]	Pt(II), phenanthridine	A2780; A2780cis; SKOV-3; MDA-MB-231; A549	Apoptosis
[31]	Pt(II), N-heterocyclic carbene (NHC)	CT26	Induction of endoplasmic reticulum stress, increase in ROS
[32]	Pt(II), benzothiazole aniline	C6; HeLa; HT-29; MCF-7	DNA intercalation
[33]	Pt(II), pyridine cationic	MCF-7; A549; Hep-G2	sterol regulatory element-binding protein 1 (SREBP-1) targeting, lipid biogenesis inhibition
[34]	Pt(II), radioactive bisphosphonate	n/a	Theragnostic, bone accumulation in mice
[35]	Pt(II), thiourea	LoVo; MCF-7	n/a
[36]	Pt(II), macrocyclic Schiff base	HeLa; A549	n/a
[37]	Pt(II), ONN-“pincer”	Hep-G2	n/a
[38]	Pt(II), 1,4-diaza-1,3-butadiene	SCOV-3	n/a
[39]	Pt(IV), axial ligands indomethacin/ acetylsalicylic acid	A2780 ADDP	n/a
[40]	Pt(IV), axial maleamide	CT-26 (in vivo murine model)	Albumin binding enhances drug accumulation in cancer cells
[41]	Pt(IV), mono- and di-axial diazido complex	A2780; A549	Photocytotoxicity, DNA binding, ROS generation
[42]	Pt(IV), diazido complex	A2780	UV-induced photocytotoxicity, ROS generation, immunogenic cell death

**Table 2 molecules-28-01959-t002:** Summary of the ruthenium complexes presented.

Citation Number	Metal Ion/Ligand Type	Effective against	Biological Activity (If Investigated)
[49]	Ru(II), p-cymene, Shiff base	Caco-2	n/a
[50,51]	Ru(II), polypyridine, 1,10-phenanthroline	SKMEL-28	Photocytotoxicity, ROS generation
[52]	Ru(II), polypyridine	B16; HepG2; A549	Disrupted cell migration, G0/G1 cell cycle arrest, ROS generation, mitochondrial membrane penetration
[53]	Ru(II), rhein-substituted polypyridine	MCF-7; A549; NB-4; A2780; A2780R	Photocytotoxicity, autophagy
[54]	Ru(II), 1,10-phenanthroline, plumbagin	MCG-803 in vivo murine model	Mitochondrial impairment, DNA damage, G0/G1 cell cycle arrest
[55]	Ru(II), polypyridine, maltol	HeLa; A2780; A2780cis; A2780ADR; CT-26; CT-26LUC	Apoptosis
[57]	Ru(II), polypyridine,	HeLa, A549	Pyroptosis, caspase 1 activation, ROS increase
[58]	Ru(II), NHC	A549; HT-29; HCT-116; LoVo; HeLa; A2780	Mitochondrial dysfunction, apoptosis, disrupted cell migration
[59]	Ru(II), cyclopentadienyl	A2780; MDAMB231; HT29	HSA binding.
[60]	Ru(II)/(III), 2-aminophenyl benzimidazole, DMSO	Caco-2, MCF-7, EAC(in vivo)	DNA-laddering, G2/M cell cycle arrest
[61]	Ru(II), dinuclear, Biginelli hybrids	HeLa; A375; K562	Inhibited cell migration and endothelial tube formation
[62]	Ru(II), p-cymene, imidazophenanthroline	HeLa; CaCo-2	CT-DNA and BSA binding
[63]	Ru(II), p-cymene, imidazophenanthroline	MDB-MA-231; HeLa	CT-DNA and BSA binding
[64]	Ru(II), diclofenac, organophosphines	A549; MDA-MB-231; MCF-7	CT-DNA and BSA binding, apoptosis
[65]	Ru(II), p-cymene, bis-aminophosphine	A375	Apoptosis
[66]	Ru(II), p-cymene, aromatic diamine	OVCAR-3; M-14; HOP-62	CT-DNA binding
[67]	Ru(II), dinuclear, p-cymene, tetrazole	HeLa; MCF-7; A549	DNA and BSA binding, G0/G1 cell cycle arrest, cell migration inhibition
[68]	Ru(II), half-sandwich arene	MCF-7	CT-DNA and BSA binding
[69]	Ru(II), flavone	MCF-7; MDA-MB-231	Cell migration inhibition
[70]	Ru(II), pyrazolopyrimidine	SCOV-3	ROS generation, mitochondrial impairment, inhibition of cell invasion and proliferation
[71]	Ru(II), aminoflavone	A2780; A2780cis; Toledo; Toledo-cis	Mitochondrial impairment, DNA interaction, apoptosis
[72]	Ru(II), dinuclear, arene	A2780; A2780cisR; A24; (D-)A24cisPt8.0	n/a
[74]	Ru(II), diphosphine, lapachol, lawsone	MDA-MB-231; MCF-7; A549; DU-145	DNA interaction. Cell cycle arrest, mitochondrial disruption, ROS increase
[75]	Ru(III), quinolone antibiotics	LoVo	G0/G1 cell cycle arrest
[76]	Ru(III), triazolopyrimidine	MCF-7, HeLa	Increased ROS generation, DNA and protein binding
[77]	Ru(III), 1,4,8,11-tetraazacyclotetradecane	SiHa	Nuclear fragmentation, apoptosis

**Table 3 molecules-28-01959-t003:** Summary of the gallium complexes presented.

Citation Number	Metal Ion/Ligand Type	Effective against	Biological Activity (If Investigated)
[9]	Ga(III), salen	HeLa, HepG 2, MCF-7, A549	Protein disulfide isomerase inhibition
[84]	Ga(III), hydrazone	MCF-7; PC-3	n/a
[90]	Ga(III), corrole	DU145; SK-MEL-28; MDA-MB-231; OVCAR-3	n/a
[91]	Ga(III), benzoylpyridine thiosemicarbazone	HepG-2	Impaired mitochondrial function, ferritin expression downregulated, transferrin receptor-1 upregulated, activated caspase-3 increased ROS
[92]	Ga(III), isonicotynoyl hydrazone	HL-60; MCF-7; HCT-116; PC3	Reduced clonogenicity.
[93]	Ga(III), polypyridine	OSC; OSCs	Damage to genomic DNA, apoptosis
[94]	Ga(III), substituted 8-quinolinol	A2780, MDA-MB-231 and HCT116	
[95]	Ga(III), 8-quinolinol	BJAB mock; Nalm-6; SK-N-AS; K652	Cell proliferation inhibition, impaired mitochondrial function, apoptosis

**Table 4 molecules-28-01959-t004:** Summary of the gold complexes presented.

Citation Number	Metal Ion/Ligand Type	Effective against	Biological Activity (If Investigated)
[102]	Au(I), 4,5-diarylimidazoles	HepG2; SMMC-7721; Hep3B	TrxR inhibition, ROS increase
[104]	Au(I), NHC	OVCAR3; NCI-H522; HT29; T-47D; PC-3	TrxR inhibition, nuclear protein Ki67 reduction
[105]	Au(I), NHC	HCT-15; A549; MCF7	n/a
[110]	Au(I), phosphine, alkynylphenanthrene	MCF-7; HEPG-2; PC-3; MOLT-4	DNA binding
[112]	Au(I)-alkynyl, triphenylphosphane	HT29; IGROV1; HL60	TrxR inhibition
[113]	Au(I)-alkynyl, triphenylphosphine/ PTA	MCF-7; HepG5; Caco-2/TC7	TrxR inhibition, COX-1/2 inhibition, increased ROS
[114]	Au(I)-alkynyl, phosphane	OVCAR-3; HOP-62	HSA binding
[115]	Au(I)-alkynyl	5637; T24	TrxR inhibition
[117]	Au(I)-alkynyl-triterpene	MCF-7; HT-29; HepG3; A2780	TrxR inhibition, mitochondrial impairment, increased ROS, endoplasmic reticulum stress
[119]	Au(I)-alkynyl-ibuprofen	MCF-7; MDA-MB-231; HT-29	TrxR and glutathione reductase inhibition, increased ROS

**Table 5 molecules-28-01959-t005:** Summary of the lanthanum complexes presented.

Citation Number	Metal Ion/Ligand Type	Effective against	Biological Activity (If Investigated)
[126]	La(III), ferrocene-substituted Schiff base	MCF-7	Molecular docking: 3HB5 breast cancer receptor
[127]	La(III), tetradentate Schiff base	MCF-7, HepG2	n/a
[128]	La(III), 5-fluorouracil	Caco2	n/a
[129]	La(III), tyrosine	MCF-7	Molecular docking: interaction with site II of BSA
[130]	La(III), polypyridine, 1,10-phenanthroline	MCF-7	DNA intercalation, DNA cleaving
[131]	La(III), bipyridyl	SK-OV-3/DDP; NCI-H460; HeLa; HL-7702	n/a
[132]	La(III), pyridine-2,6-dicarboxylate	HL60; HepG2; HT29	ROS increase
[133]	La(III), bipyridine	MCF-7; A-549	Molecular docking: interaction with site III of BSA

## Data Availability

Not applicable.

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
