# Peer review of "Recent Trends in the Development of Novel Metal-Based Antineoplastic Drugs"

_molecules, 2023, doi:10.3390/molecules28041959_

Round 1
Reviewer 1 Report
I found the review really necessary to shed some light in the very complicated topic of novel metal prodrugs discovery, that should connect distinct expertise in inorganic, organometallic chemistry, biochemical mechanisms with pharmaceutical knowledges and the oncology-medicine consciousness. The ratio of the review is oriented towards an holistic approach. The result is "soundy" and amenably written. The connections between structures, variations of ligands and the related bio-activity applications are clearly reported and both scheduled with the relative references in very useful summary-tables. The descriptions of the biochemical paths are described in a such an attractive mode to make able to easy understand even no-specialists in the field. The key and the hearth of the matter are constantly highlighted, with the idea of making the reader aware about the global, multifaceted platform to rationally design novel drugs. And with no aid of Artificial Intelligence.
Minor alterations and miswritings:
118 antiproliferative
253 binds
279 I would suggest.. rate change for Ru(II) and Pt(II). 280 Ru(III) goes to replace Pt (III) at the end of the sentence
387 induced
405-406 2.5 instead of 2,5 673 essay
676 I suggest .. were about 2-3 times higher than those for cisplatin.
720 lower
Reviewer 2 Report
The manuscript entitled “Recent Trends in the Development of Novel Metal-Based Antineoplastic Drugs” by Lozan Todorov, and Irena Kostova presents an interesting review of the recent literature (covering the las three years) of Pt, Ru, Ga, Au and La. This work will be useful for researchers in the field.
Despite that, it requires to be improved before it is ready for publication. In general, it deserves a detailed revision by the authors to fix different problems in its redaction. Being a review, a more systematic presentation is advisable. For instance, it would be adequate to use a numbering system for the complexes (at least for each metal). The IC50 is not always present, it would be advisable to follow whenever possible a similar structure for the description of each work which includes the IC 50 , the cell lines in which it was determined and the test duration. In addition, it is necessary to revise the abbreviation definition, some are defined several times (for instance BSA), others are not defined the first time they are used.
In addition, I have copied some parts that I found misleading or with non-adequate content to fix, which are attached.

Reviewer 3 Report
The review of Lozan Todorov and Irena Kostova is about metal-based coordination compounds with biological properties. The review has a lot of good quality chem schemes which help to navigate and understand manuscript.
The author made a really good job. Despite the review doesn’t have critical information, advices to early career researchers and trends it has a lot of general information about metallodrugs. I understand author’s choice to publish the review without pay-wall in open access journal so that as many people as possible can read the above.
Line 15: “Coordination compounds of gold (Au) have proven their effective diagnosis and/or treatment of oncological diseases” - Is there really a commercial analogue of cisplatin based on a gold ion?
Line 17: “has been gaining traction” or attraction?
Line 38: “Pb(II)/(IV)” – Platinum
Line 42: “Pb, Ru, Au and La” – Platinum
From the chapter on platinum, it is not clear which platinum (II or IV) is more effective? It might be worth making a conclusion at the end of the chapter. It is also not entirely clear whether coordination of Pt ion needs to preserve the environment, as in cisplatin (two nitrogen atoms and two chlorine atoms). And what type of ligands would the authors recommend paying attention to?
Also, I believe, if the authors deem it appropriate, it would be worth describing the following works:
10.1021/acs.inorgchem.1c03314 “Heteroleptic Pd(II) and Pt(II) Complexes with Redox-Active Ligands: Synthesis, Structure, and Multimodal Anticancer Mechanism”
10.3390/molecules27238565 “Diimine Cisplatin Derivatives: Synthesis, Structure, Cyclic Voltammetry and Cytotoxicity”
Line 310: What is a EC50 and what the difference between EC50 and IC50? How does ruthenium affect EC50 of the ligands (what the mechanism)?
Based on the chapter on ruthenium, we can conclude that the most studied is divalent ruthenium. While the efficiency of the trivalent is no less. How can this be explained? Maybe working with trivalent ruthenium is more difficult? And is it worth paying attention to this? Or is it not worth it, since trivalent ruthenium can be formed in the body under the action of oxidases? And a similar question as for platinum: what ligands should young researchers pay attention to? Maybe compounds with a ruthenium-carbon bond?
The chapter on gallium also lacks an intermediate conclusion. From the lists of works, it is not entirely clear what to pay attention to, which ligand to choose.
For gold compounds with biological activity, the current trend is the metal-carbon bond, which the authors demonstrated in the current review. What can be said about the stability of such organometallic complexes in biological media? What is the mechanism of action of the gold-carbon bond, why is it so important?
In any case, the presented review is an interesting work and I certainly recommend accepting it for publication in the journal Molecules. This work will definitely attract a lot of attention from researchers and increase citations for the Molecules.
Round 2
Reviewer 2 Report
The manuscript has improved its organization and redaction and is adequate for publication.
There are only some problems with Figs, 7, 11, 24, 37 that are not being displayed.